# LoLA: Low-Rank Linear Attention with Sparse Caching

**Luke McDermott**                                                          *lmcdermo@ucsd.edu*
*University of California, San Diego*

**Robert W. Heath Jr.**                                                      *rwheathjr@ucsd.edu*
*University of California, San Diego*

**Rahul Parhi**                                                             *rahul@ucsd.edu*
*University of California, San Diego*

**Reviewed on OpenReview:** *https://openreview.net/forum?id=3KhDA252y3*

## Abstract

The per-token cost of transformer inference scales with context length, preventing its application to lifelong in-context learning. Linear attention is an efficient alternative that maintains a constant memory footprint, even on infinite context lengths. While this is a potential candidate for lifelong learning, it falls short in memory capacity. In this paper, we propose LoLA, a training-free augmentation to linear attention that boosts associative recall. LoLA distributes past key-value pairs from context into three memory systems: (i) recent pairs in a local sliding window cache; (ii) difficult-to-memorize pairs in a sparse, global cache; and (iii) generic pairs in the recurrent hidden state of linear attention. We show through ablations that our self-recall error metric is crucial to efficiently manage long-term associative memories. On pass-key retrieval tasks, LoLA improves the base model's performance from 0.6% to 97.4% accuracy. This is achieved with a $4.6\times$ smaller cache than Llama-3.1 8B on 4K context length. LoLA also outperforms other 1B and 8B parameter subquadratic models on zero-shot commonsense reasoning tasks.

## 1 Introduction

Transformer-based large language models (LLMs) rely on storing all past tokens in an ever-growing key-value (KV) cache (Vaswani et al., 2017). This allows future query tokens to access past memories with associative recall, which enables in-context learning (Olsson et al., 2022). Since no previous information is discarded, the KV cache continues to grow with context length. This eventually leads to a memory bottleneck on long context tasks, such as lifelong in-context learning. As a result, transformers cannot condition next token predictions on arbitrarily long sequences.

Alternative architectures to transformers have been proposed—such as Mamba (Gu & Dao, 2024), DeltaNet (Schlag et al., 2021), linear attention (Katharopoulos et al., 2020), and others (Yang et al., 2024a; Behrouz et al., 2024; Sun et al., 2024)—to reduce the compute complexity from quadratic to linear. Additionally, these approaches reduce the memory cost from linear to constant. In particular, linear attention removes the exponential dot product in softmax (Katharopoulos et al., 2020). This effectively collapses the unbounded KV-cache into a fixed-size matrix, which corresponds to a recurrently formed hidden state (i.e., a linear RNN). This constructs a linear associative memory map from keys to values. Past memories can be recalled through a vector-matrix product of an incoming query vector and the hidden state matrix. Linear attention enables constant-cost prediction per token when conditioned on arbitrarily long contexts.

While efficient and flexible, linear attention architectures lag behind transformers in terms of memory capacity. This is largely noticeable on tasks leveraging in-context learning (Paperno et al., 2016; Hsieh et al., 2024). The removal of the exponential dot product allows for non-orthogonal keys to interfere with the hidden state's learned key-to-value map. This interference—denoted as a *memory collision*—impairs associative

recall. Despite linear attention's efficiency and simplicity, this limitation in memory capacity prevents more widespread adoption. Previous work used nonlinear query and key activations to improve the exponential dot product approximation (Choromanski et al., 2020; Zhang et al., 2024). However, these attempts are essentially performing a low-rank approximation of the infinite-rank exponential dot product kernel.

Additional use of sparse attention (Chen et al., 2021) can improve linear attention's recall; however, hybrid approaches often only focus on local information with sliding window attention (Arora et al., 2024; Zhang et al., 2025a; Lan et al., 2025; Van Nguyen et al., 2025). These approaches can struggle to recall critical, long-term facts that fall outside the window.

This raises our fundamental research question:

*How can long term associative memory for subquadratic language models be improved?*

**Contributions**

We present LoLA: Low-rank Linear Attention with sparse caching. LoLA is a novel inference augmentation that boosts the performance of hybrid linear attention layers. LoLA distributes historical tokens into three forms of memory: (i) recent KV pairs are stored in a sliding window cache, (ii) difficult-to-memorize pairs in a sparse global cache, and (iii) all other pairs are placed in a recurrent hidden-state matrix via linear attention. LoLA performs a self-recall check to see which KV pairs disagree with the current hidden state's linear associative map. LoLA sparsely caches the interfering memories in full rank. The selection mechanism effectively mitigates memory collisions with a small, constant-sized cache. This inference strategy can be applied on top of previously trained linear attention + sliding window models (e.g., LoLCATs) to significantly improve associative recall. As a result, LoLA extracts stronger language modeling capabilities from the same base model weights.

**Utilizes Self-Recall Error.** We introduce an importance metric for key-value pairs to reduce memory collisions in the hidden state of linear attention. This is computed by determining if a key can recall its own value with linear attention. In our ablations, we show that this performance increase cannot be obtained from using a larger sliding window or other sparse attention metrics.

**Enables Associative Recall.** As a lightweight inference strategy, LoLA enables pass-key retrieval on up to 8K context lengths in needle-in-a-haystack tasks from the RULER benchmark (Hsieh et al., 2024). With a **4.6x smaller** cache than Llama-3.1 8B (Grattafiori et al., 2024), our approach boosts accuracy from LoLCATs' 0.6% to 97.4% at 4K context lengths with the same model weights.

**Improves Language Modeling.** LoLA shows superior performance on zero-shot commonsense reasoning tasks among 1B and 8B parameter subquadratic architectures. This demonstrates that effective memory management can boost language modeling performance.

## 2 Preliminaries

In this section, we review softmax attention through the lens of associative memory. Then, we show how linear attention naturally forms a recurrent hidden state. We address practical implementations for training linear architectures and highlight unresolved drawbacks of previous approaches.

### 2.1 Softmax Attention as a Nonparametric, Online Learner

Transformers process a sequence of input tokens $\{\boldsymbol{x}_t\}_{t=1}^n$, for $\boldsymbol{x}_t \in \mathbb{R}^d$ (Vaswani et al., 2017). For each attention head, the input tokens are transformed into three distinct representations—queries, keys, and values—via trainable weight matrices $\mathbf{W}_q, \mathbf{W}_k \in \mathbb{R}^{d \times d_k}$ and $\mathbf{W}_v \in \mathbb{R}^{d \times d_v}$. For a given token $\boldsymbol{x}_t$, define

$$\underbrace{\boldsymbol{q}_t = \mathbf{W}_q \boldsymbol{x}_t}_{\text{query}}, \quad \underbrace{\boldsymbol{k}_t = \mathbf{W}_k \boldsymbol{x}_t}_{\text{key}}, \quad \underbrace{\boldsymbol{v}_t = \mathbf{W}_v \boldsymbol{x}_t}_{\text{value}}. \tag{1}$$

Causal attention uses the current query to recall past information from key-value pairs. The similarity between the query $\boldsymbol{q}_t$ and key $\boldsymbol{k}_i$ is denoted as $\alpha_{ti} \in (0,1)$. This similarity score determines how much value $\boldsymbol{v}_i$ is used for the current output token at time $t$. The output token $\boldsymbol{y}_t$ is defined by

$$\boldsymbol{y}_t = \sum_{i=1}^{t} \alpha_{ti}\,\boldsymbol{v}_i \in \mathbb{R}^{d_v}, \quad \text{with} \quad \alpha_{ti} = \frac{\exp\left(\boldsymbol{q}_t^\top \boldsymbol{k}_i / \sqrt{d_k}\right)}{\sum_{j=1}^{t} \exp\left(\boldsymbol{q}_t^\top \boldsymbol{k}_j / \sqrt{d_k}\right)}. \tag{2}$$

We view softmax attention as a nonparametric function $m_t : \mathbb{R}^{d_k} \to \mathbb{R}^{d_v}$ that fits to the past context at inference time. This online function $m_t$ learns to map keys to their associated value in context with $m_t(\boldsymbol{k}_i) \approx \boldsymbol{v}_i$ for $(\boldsymbol{k}_i, \boldsymbol{v}_i) \in \{(\boldsymbol{k}_i, \boldsymbol{v}_i)\}_{i=1}^{t}$. Then, $m_t$ applies the learned transformation to the query, $\boldsymbol{y}_t = m_t(\boldsymbol{q}_t)$. The set of past key-value pairs forms an online "training set" of input-output labels. The query acts as an unsupervised "test set".

Softmax attention caches all past key-value pairs to perform this non-parametric, or "look-up table", operation. Since the function complexity scales with the context length, softmax attention can flexibly learn new context without forgetting past key-value associations. However, this process leads to an unbounded KV-cache size that scales linearly with sequence length $n$. Ultimately, this operation cannot be used for extremely long context scenarios, such as lifelong learning.

## 2.2 Linear Attention

To bound inference costs, linear attention methods replace the exponential dot product kernel (Katharopoulos et al., 2020) with a low-rank approximation. This enables models to maintain constant-size memory footprints even for infinite sequence lengths. With this replacement,

$$\exp\left(\frac{\boldsymbol{q}_t^\top \boldsymbol{k}_j}{\sqrt{d_k}}\right) \approx \phi(\boldsymbol{q}_t)^\top \phi(\boldsymbol{k}_j), \quad \text{for} \quad \phi : \mathbb{R}^{d_k} \to \mathbb{R}^D, \tag{3}$$

the output token is approximated as

$$\boldsymbol{y}_t^\top = \sum_{i=1}^{t} \frac{\exp\left(\boldsymbol{q}_t^\top \boldsymbol{k}_i / \sqrt{d_k}\right)\,\boldsymbol{v}_i^\top}{\sum_{j=1}^{t} \exp\left(\boldsymbol{q}_t^\top \boldsymbol{k}_j / \sqrt{d_k}\right)} \approx \frac{\phi(\boldsymbol{q}_t)^\top \left(\sum_{j=1}^{t} \phi(\boldsymbol{k}_j)\,\boldsymbol{v}_j^\top\right)}{\phi(\boldsymbol{q}_t)^\top \left(\sum_{j=1}^{t} \phi(\boldsymbol{k}_j)\right)} = \frac{\phi(\boldsymbol{q}_t)^\top \mathbf{H}_t}{\phi(\boldsymbol{q}_t)^\top \boldsymbol{s}_t}. \tag{4}$$

This creates a hidden state matrix $\mathbf{H}_t \in \mathbb{R}^{D \times d_v}$ as the sum of key-value outer products. This effectively bounds the memory cost to $\mathcal{O}(Dd_v)$, constant with respect to sequence length $n$. The hidden dimension size $D$ controls the approximation quality at the cost of computational efficiency. As a linear RNN, the hidden state $\mathbf{H}_t$ and normalization state $\boldsymbol{s}_t \in \mathbb{R}^D$ can be computed recurrently,

$$\mathbf{H}_t = \mathbf{H}_{t-1} + \phi(\boldsymbol{k}_t)\,\boldsymbol{v}_t^\top, \quad \boldsymbol{s}_t = \boldsymbol{s}_{t-1} + \phi(\boldsymbol{k}_t). \tag{5}$$

In this formulation, linear attention stores each observation, or KV-pair, as a rank-one outer product. Rather than building a look-up table, linear attention parameterizes the key-to-value map as a linear function. While this approach is efficient, linear attention falls short in memory capacity as the number of orthogonal key-value pairs is bounded by the rank of $\mathbf{H}_t$. "Memory collisions" (Yang et al., 2024a) occur when new hidden state updates overwrite past key-value associations. This prevents the linear map from accurately modeling the context.

## 2.3 Efficient Training of Linear Attention

To reduce training costs, LoLCATs (Zhang et al., 2025a) and others (Bick et al., 2024; Wang et al., 2024; Bick et al., 2025; Goldstein et al., 2025) recycle large pretrained transformers into linear attention models with knowledge distillation (Hinton et al., 2015). These approaches minimize the difference between the pretrained transformer's output $\boldsymbol{y}$ (i.e., the teacher) and linear attention's output $\widehat{\boldsymbol{y}}$ (i.e., the student). In particular, LoLCATs uses a trainable nonlinear map for $\phi : \mathbb{R}^{d_k} \to \mathbb{R}^D$, constructed as

$$\phi(\boldsymbol{x}) = \left[\exp(\boldsymbol{w}_1^\top \boldsymbol{x}), \ldots, \exp(\boldsymbol{w}_{D/2}^\top \boldsymbol{x}), \exp(-\boldsymbol{w}_1^\top \boldsymbol{x}), \ldots, \exp(-\boldsymbol{w}_{D/2}^\top \boldsymbol{x})\right] \in \mathbb{R}^D, \tag{6}$$

with learnable weights $\boldsymbol{w}_i \in \mathbb{R}^{d_k}$ (Zhang et al., 2024). This distillation approach freezes all other parameters, adjusting $\phi$ to minimize the loss

$$\mathcal{L}(\phi) = \sum_{\boldsymbol{q}, \boldsymbol{k}, \boldsymbol{v}} \|\boldsymbol{y}_t - \widehat{\boldsymbol{y}}_t\|, \quad \text{with} \quad \widehat{\boldsymbol{y}}_t = \frac{\phi(\boldsymbol{q}_t)^\top \mathbf{H}_t}{\phi(\boldsymbol{q}_t)^\top \boldsymbol{s}_t}. \tag{7}$$

After attention distillation, the whole model is finetuned with LoRA (Hu et al., 2022). Overall, this procedure only requires 40 million training tokens from the Alpaca dataset (Taori et al., 2023), grouped in 1024-long sequences. While efficient, this distillation does not transfer all knowledge. In particular, the distilled model has not been trained to store context beyond 1024 tokens.

## 2.4 Disadvantages of Previous Approaches

Even with distillation, linear attention models struggle to accurately mimic the behavior of softmax attention. Initial work in linear attention proposed nonlinear query and key activations to improve the exponential dot product approximation (Choromanski et al., 2020; Zhang et al., 2024). These methods fall short as they are essentially performing a low-rank approximation of the infinite-rank exponential dot product kernel. In Appendix E, we show that the exponential dot product kernel has slowly decaying singular values for simple data distributions. This implies that high-dimensional hidden states may be required for modest approximation errors.

Recent approaches attempt to address the poor performance of the low-rank approximation in linear attention by augmenting it with sliding window attention (Arora et al., 2024; Zhang et al., 2025a). These methods compute a *finite* number of recent tokens in a window with softmax attention and compute the rest with linear attention. Since natural language contains a significant amount of local information, this hybrid approach nearly recovers the performance of pretrained transformers on short-context tasks. For longer sequences, however, these methods struggle to recall important information that falls outside the window and in the hidden state. We show in Table 1 that these models cannot perform associative recall on simple needle-in-a-haystack tasks. Other forms of sparse attention may be required alongside "low-rank" attention (Chen et al., 2021).

# 3 Mitigating Memory Collisions with Sparse Caching

**Identifying Difficult-to-Remember KV Pairs.** As our base assumption, strong associative memory systems should allow keys to retrieve their associated values. Online functions $m_t$ with perfect recall interpolate the online training set, defined as

$$m_t(\boldsymbol{k}_i) = \boldsymbol{v}_i, \quad \forall i \leq t. \tag{8}$$

For perfect recall in linear attention, equation 8 translates to

$$m_t(\boldsymbol{k}_i)^\top = \frac{\phi(\boldsymbol{k}_i)^\top \mathbf{H}_t}{\phi(\boldsymbol{k}_i)^\top \boldsymbol{s}_t} = \frac{\phi(\boldsymbol{k}_i)^\top \sum_{j=1}^t \phi(\boldsymbol{k}_j) \boldsymbol{v}_j^\top}{\phi(\boldsymbol{k}_i)^\top \sum_{j=1}^t \phi(\boldsymbol{k}_j)} = \boldsymbol{v}_i^\top. \tag{9}$$

In practice, however, "memory collisions" prohibit equation 9 from holding. Non-orthogonal keys interfere with each other when forming the hidden state $\mathbf{H}_t$. We measure the Self-Recall Error (SRE) to assess how well a past key can retrieve its associated value with

$$\text{SRE}(\boldsymbol{k}, \boldsymbol{v} \mid \mathbf{H}_t, \boldsymbol{s}_t) = \|m_t(k) - \boldsymbol{v}\|_2 = \left\| \frac{\phi(\boldsymbol{k})^\top \mathbf{H}_t}{\phi(\boldsymbol{k})^\top \boldsymbol{s}_t} - \boldsymbol{v} \right\|_2 = \|\hat{\boldsymbol{v}} - \boldsymbol{v}\|_2. \tag{10}$$

This determines the error between the predicted value $\hat{\boldsymbol{v}}$ for a given key $\boldsymbol{k}$ and the ground truth value $\boldsymbol{v}$. In our particular implementation, we leverage linear attention with feature maps $\phi$; however, the SRE can be used to measure memory corruption for any $m_t$, such as other test-time training methods (Behrouz et al., 2025; Zhang et al., 2025b; Yang et al., 2024a)

**Method Overview.** We propose LoLA: a novel inference strategy that boosts the performance of hybrid linear attention models. LoLA addresses the limitations of previous linear attention mechanisms by integrating a sparse caching strategy at inference time. As illustrated in Figure 1, this method employs three memory systems to store long term associations

1. **Linear Attention** utilizes a finite-rank approximation to store an infinite amount of tokens.

2. **Sliding Window Attention** provides full-rank attention scores for finite, local context.

3. **Sparse Caching** identifies and stores key-value pairs that are challenging to remember, preventing memory collisions in linear attention.

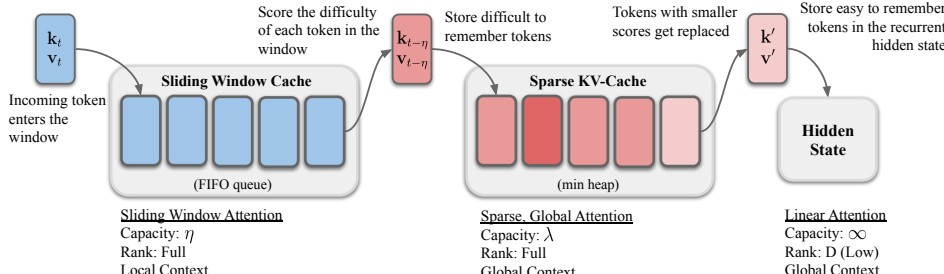

Figure 1: LoLA stores past KV pairs in three forms of memory for each attention head.

LoLA uses the self-recall error, equation 10, to decide which KV pairs should be stored separately in full-rank. Large errors indicate the severity of the memory collision. As a result, LoLA keeps the KV pairs with the largest error in a sparse cache. This limits the corruption of past memories and improves associative recall. Additionally, with our chunkwise implementation, LoLA only updates the sparse cache after each chunk of tokens, dictated by the sliding window size.

Since only a finite amount of tokens can be stored at each time step to maintain efficiency, LoLA performs a greedy scoring approach. At every update interval, LoLA scores the KV pairs leaving the sliding window and re-scores all pairs currently stored in the sparse cache. During each update, the pairs with the lowest error are moved to the linear hidden state indefinitely. Re-scoring pairs in the sparse cache is essential since the SRE is dependent on the current hidden state. For example, a KV pair could become more aligned with the hidden state in the future after a few updates. In the generation implementation of LoLA, we define the set of pairs that are scored at time $t$ as

$$\mathcal{E}_t = G_{t-1} \cup \{(\boldsymbol{k}_{t-\eta}, \boldsymbol{v}_{t-\eta})\}, \tag{11}$$

where $G_{t-1}$ is the set of KV pairs in the sparse cache at time $t-1$. Here, $\eta$ is the maximum number of pairs in the sliding window. We update the sparse cache by selecting the top-$\lambda$ errors in $\mathcal{E}_t$, i.e,

$$G_t = \underset{G \subset \mathcal{E}_t : |G| = \lambda}{\arg\max} \sum_{(\boldsymbol{k}, \boldsymbol{v}) \in G} \text{SRE}(\boldsymbol{k}, \boldsymbol{v} \mid \mathbf{H}_t, \boldsymbol{s}_t). \tag{12}$$

The remaining pairs, denoted by $\mathcal{S}_t = \mathcal{E}_t \cap G_t^{\mathsf{c}}$ where $G_t^{\mathsf{c}}$ is the complement of $G_t$, are stored in hidden state via

$$\mathbf{H}_t = \mathbf{H}_{t-1} + \sum_{(\boldsymbol{k}, \boldsymbol{v}) \in \mathcal{S}_t} \phi(\boldsymbol{k}) \boldsymbol{v}^\top, \quad \boldsymbol{s}_t = \boldsymbol{s}_{t-1} + \sum_{(\boldsymbol{k}, \boldsymbol{v}) \in \mathcal{S}_t} \phi(\boldsymbol{k}). \tag{13}$$

Once the caches are up to date, LoLA computes the output token $\boldsymbol{y}_t$ as

$$\boldsymbol{y}_t = \frac{\overbrace{\phi(\boldsymbol{q}_t)^\top \mathbf{H}_t}^{\text{Linear Attn.}} + \overbrace{\sum_{i \in G_t} \exp\left(\boldsymbol{q}_t^\top \boldsymbol{k}_i / \sqrt{d_k}\right) \boldsymbol{v}_i}^{\text{Sparse Cache}} + \overbrace{\sum_{j=t-\eta+1}^{t} \exp\left(\boldsymbol{q}_t^\top \boldsymbol{k}_j / \sqrt{d_k}\right) \boldsymbol{v}_j}^{\text{Sliding Window}}}{\phi(\boldsymbol{q}_t)^\top \boldsymbol{s}_t + \sum_{i \in G_t} \exp\left(\boldsymbol{q}_t^\top \boldsymbol{k}_i / \sqrt{d_k}\right) + \sum_{j=t-\eta+1}^{t} \exp\left(\boldsymbol{q}_t^\top \boldsymbol{k}_j / \sqrt{d_k}\right)}. \tag{14}$$

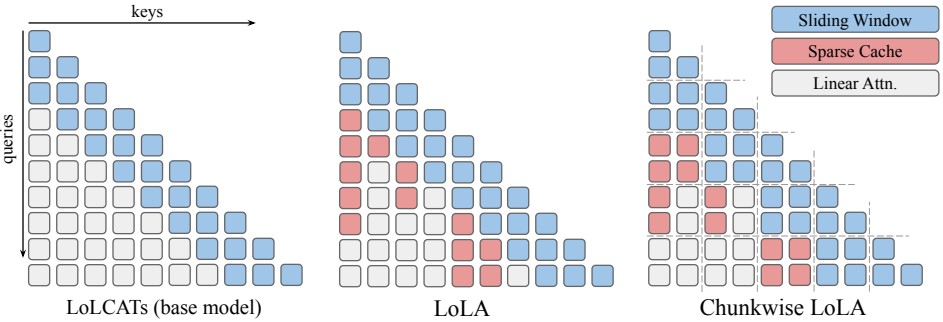

Figure 2: Illustration of where each KV pair is stored at every time step for each method.

The parameter $\eta$ sets the sliding window size, and $\lambda$ sets the sparse cache size.

**Chunkwise Inference.** When the input sequence is available ahead of time (e.g., prefill), LoLA is accelerated with parallelization. By partitioning the input sequence into chunks of size $C$, we can compute intra-chunk operations in parallel with dense matmuls (Yang et al., 2024c). This reduces the number of recurrent iterations by a factor of $C$ while preserving the constant-memory cost that motivates LoLA.

LoLA computes softmax attention with the current chunk of queries and previous two chunks of KV-pairs. For processing a small chunk, softmax attention is almost equally efficient to linear attention. Artificially limiting softmax within a chunk will not improve efficiency, only hurt performance. The past two chunks of KV-pairs (the sliding window) are concatenated with the sparse cache in order to compute a single FlashAttention (Dao et al., 2022) pass per chunk of queries. For the linear attention portion of the forward pass, all queries within the chunk share the same hidden state.

After computing the past chunk of output tokens, we evict the oldest chunk of KV-pairs in the window, sending them to the hidden state or sparse cache. All of the evicted and sparse cache pairs are scored with the SRE, equation 10. The $\lambda$ pairs with the largest errors in the eligible set,

$$\mathcal{E}_t = G_{t-1} \cup \{(\mathbf{k}_i, \mathbf{v}_i) \mid t - 2C \leq i < t - C\}, \tag{15}$$

will remain in the sparse cache. The remainder are integrated into the hidden state through the standard outer product update as in equation 5. We illustrate this chunkwise process in Figure 2.

The cache parameters $\eta$ and $\lambda$ present a trade-off between latency and performance. The given hardware setup dictates the total fixed cache size of LoLA, but the ratio of chunk size to sparse cache size depends on the application. Increasing the chunk or sliding window size reduces the number of recurrent iterations (i.e. number of cache updates), thus decreasing the overhead costs from sparse caching. However, this requires more VRAM. Increasing the sparse cache size $\lambda$ will better mitigate collisions in the hidden state and improve long context recall. Furthermore, for moderate chunk sizes, the update frequency slows, reducing the relative impact of large sparse caches. In Appendix B, we explore this trade-off for various cache hyperparameters in an efficiency analysis. Specifically, we measure the total VRAM use, Time-to-First-Token, and long context performance. Furthermore, we illustrate the bounded nature of LoLA, compared to vanilla transformers.

## 4 Experiments & Results

For our experiments, we leverage LoLCATs' efficient distillation procedure to create the base model. Then, we apply our inference strategy, LoLA, at test time. Distillation is not required for LoLA, only a trained intra-head SWA+LA model. To train the base model, we replace each attention module in Llama-3.1 8B (or Llama-3.2 1B) with a hybrid sliding window + linear attention module. We use a sliding window size $\eta = 64$ for training and use a trainable feature map for $\phi$ as described in equation 6. The output dimension of $\phi$ is $D = 2d_k$. First, we freeze all non-attention layers in the linearized Transformer and only train $\phi$ with distillation for two epochs on Alpaca (Taori et al., 2023). Then, we perform LoRA (Hu et al., 2022)

finetuning on the whole model for two epochs. This procedure only uses 40M training tokens with 1024-long sequences.

## 4.1 Associative Recall

To see how LoLA improves associative recall, we conduct a study on Single-Needle-in-a-Haystack (S-NIAH) tasks from RULER (Hsieh et al., 2024). In Table 1, we compare LoLA to the base model, LoLCATs-8B, and variants with an extended sliding window for a fair comparison. We observe LoLCATs struggles to recall information outside the sliding window. Extending the sliding window size marginally improves performance. We explain the differences of each NIAH task in Appendix C and discuss these results more in depth.

Table 1: Measuring long context recall with Needle-in-a-Haystack (NIAH) tasks from the RULER benchmark. We report recall accuracy for each method across different context lengths (512, 1024, etc.) for each task. For a stronger baseline, we extend LoLCATs to use a larger cache size, denoted with "+"; though, this is not used in (Zhang et al., 2025a).

| Model | Cache Params $(\eta, \lambda)$ | S-NIAH-1 | | | | S-NIAH-2 | | | S-NIAH-3 | | |
|---|---|---|---|---|---|---|---|---|---|---|---|
| | | .5K | 1K | 2K | 4K | .5K | 1K | 2K | .5K | 1K | 2K |
| **Transformer** | | | | | | | | | | | |
| Llama-3.1-8B | $(\infty, 0)$ | 100 | 100 | 100 | 100 | 100 | 100 | 100 | 100 | 100 | 100 |
| **Base Subquadratic Model** | | | | | | | | | | | |
| LoLCATs-8B | $(64, 0)$ | 9.0 | 3.2 | 1.4 | 0.6 | 100 | 7.6 | 2.0 | 97.4 | 1.6 | 0.6 |
| **Extended at Inference** | | | | | | | | | | | |
| LoLCATs-8B+ | $(128, 0)$ | 29.4 | 9.6 | 3.4 | 1.4 | 100 | 17.4 | **7.2** | 98.2 | **14.6** | **3.2** |
| LoLA-8B | $(64, 64)$ | **99.0** | **95.4** | **79.4** | **69.4** | 100 | **39.4** | 3.0 | **99.8** | 7.4 | 1.6 |
| LoLCATs-8B+ | $(512, 0)$ | 100 | 65.6 | 24.6 | 8.8 | 100 | 71.8 | 21.6 | **100** | 66.0 | 10.6 |
| LoLA-8B | $(256, 256)$ | 100 | **100** | **99.6** | **97.4** | 100 | **100** | **85.4** | 99.8 | **99.8** | **27.2** |
| LoLCATs-8B+ | $(1024, 0)$ | 100 | 100 | 55.4 | 30.4 | 100 | 100 | 60.0 | 100 | 100 | 27.6 |
| LoLA-8B | $(512, 512)$ | 100 | 100 | **100** | **100** | 100 | 100 | **100** | 100 | 100 | **100** |

Next in Table 2, we measure the performance of LoLA on the rest of the RULER benchmark at 4K context length. This covers much harder long context tasks, such as multi-key retrieval (MK1,MK2,MK3), multi-query (MQ), multi-value (MV), variable tracking (VT), common word extraction (CWE), frequent word extraction (FWE), Hotpot-QA (HQA), and Squad-QA (SQA). With these harder tasks, we increase sparse cache size to $\lambda = 768$ and reduce the sliding window size to $\eta = 128$. We compare LoLA against a stronger version of LoLCATs—with an equivalent, larger cache size ($\eta = 896$)—and Mamba2-8B (Dao & Gu, 2024; Waleffe et al., 2024).

Table 2: Extended RULER Benchmark on 4K context lengths. Compared to NIAH tasks, these require much stronger forms of memory and state tracking. For cache parameters, LoLA uses $\eta = 128, \lambda = 768$ and LoLCATs+ uses $\eta = 896$. Llama-3.1 performance is averaged across 50 samples per task.

| Model | MK1 | MK2 | MK3 | MQ | MV | VT | CWE | FWE | HQA | SQA | Avg |
|---|---|---|---|---|---|---|---|---|---|---|---|
| **Transformer** | | | | | | | | | | | |
| Llama-3.1-8B | 100 | 100 | 98 | 90.5 | 98.5 | 99.6 | 95.2 | 91.7 | 62 | 81.5 | 91.7 |
| **Subquadratic** | | | | | | | | | | | |
| Mamba2-8B | **40.3** | **13.8** | 5.5 | 49.1 | 35.0 | 76.5 | 32.9 | **76.6** | **31.8** | 35.5 | 39.7 |
| LoLCATs-8B+ | 12.8 | 1.4 | 0.4 | 3.3 | 3.6 | 0.7 | 3.0 | 13.6 | 14.2 | 14.0 | 6.7 |
| LoLA-8B | 39.4 | 11.6 | **7.6** | **67.6** | **65.0** | **85.2** | **45.9** | 51.3 | 24.2 | **53.9** | **45.2** |

LoLA improves recall with minimal additional caching. Table 1 demonstrates an improvement from the base model's 0.6% to 97.4% accuracy on S-NIAH-1. This is achieved with a 4.6× smaller cache than Llama with

$\eta = 256, \lambda = 256$. Furthermore, we show in Table 2 that sparse caching enables long-context understanding for more difficult tasks, improving an extended form of LoLCATs from 6.7% average accuracy to 45.2%. For example, tasks such as variable tracking (VT) require understanding all of the context. Since no part of the sequence can be lost, naive metrics for sparse attention, such as (Zhang et al., 2023), cannot be used. Memory collisions must be mitigated. Sparse caching—specifically with our self-recall error—unlocks a new capability for hybrid linear attention architectures.

## 4.2 Commonsense Reasoning

We demonstrate the language modeling performance of LoLA on various zero-shot commonsense reasoning tasks using LM eval harness (Gao et al., 2024). To compare against previously available approaches, we use PIQA (PI) (Bisk et al., 2020), ARC-Easy (AE) & ARC-Challenge (AC) (Clark et al., 2018), HellaSwag (HS) (Zellers et al., 2019), WinoGrande (WG) (Sakaguchi et al., 2021), MMLU (MM) (Hendrycks et al., 2020), and Lambada OpenAI (Paperno et al., 2016)).

In Table 3, we compare LoLA against other 7-9B *subquadratic* models [1]. Models that use any form of unbounded global attention (e.g., interleaving SSM blocks and Transformer blocks (Chen et al., 2026)) still retain quadratic compute complexity and growing memory costs. These are outside the scope of this work. We also report the number of training tokens used to create each model in both tables. Though LoLA is an inference-time strategy that can be used for any sliding window + linear attention model, we report the cost of distilling the base subquadratic model (Zhang et al., 2025a). In Appendix A, we show results for 1-2B subquadratic models. Additionally, we provide a direct comparison of distilled models by measuring the average accuracy relative to their teacher models.

Table 3: Performance comparison of 7-9B parameter fixed-memory models across various commonsense reasoning tasks: PIQA (PI), Arc-Easy (AE), ARC-Challenge (AC), Winogrande (WG), MMLU (MM), and Lambada-openai (LB). Bolded scores are the best and underlined scores are the second best. We report accuracy for all applicable, except AC and HS use normalized logits. MMLU (MM) uses 5-shot. Reported scores were compiled from (Bick et al., 2025; Zhang et al., 2025a; Waleffe et al., 2024; Wang et al., 2024). * indicates our reproduced score is used and is higher than reported score. LoLA uses small cache parameters, $\eta = \lambda = 64$.

| Model | Tokens (B) | PI | AE | AC | HS | WG | MM | LB |
|---|---|---|---|---|---|---|---|---|
| **Transformers** | | | | | | | | |
| Llama-3.1-8B | 15000 | 81.1 | 81.7 | 55.1 | 79.3 | **73.9** | **68.0** | 73.0 |
| **Subquadratic:** *Pretrained from scratch* | | | | | | | | |
| Mamba-8B | 1100 | 78.9 | 75.4 | 42.2 | 75.6 | 68.3 | 28.0 | - |
| Mamba2-8B | 3500 | 79.8 | 75.9 | 48.1 | 77.7 | 71.6 | 48.7 | - |
| RWKV-6 (W2.1) 7B | 1420 | 78.7 | 76.8 | 46.3 | 75.1 | 70.0 | - | - |
| Hawk 7B | 300 | 80.0 | 74.4 | 45.9 | 77.6 | 69.9 | 35.0 | - |
| Griffin 7B | 300 | 81.0 | 75.4 | 47.9 | 78.6 | 72.6 | 39.3 | - |
| Falcon3-Mamba-7B | 7300 | 79.7 | 72.5 | 53.2 | 79.8 | 69.1 | 65.0 | 67.5 |
| RecurrentGemma-9B | 2000 | 80.6 | 78.9 | **57.1** | **80.1** | 73.7 | 55.1 | 54.1 |
| **Subquadratic:** *Distilled from Llama-3.1-8B* | | | | | | | | |
| Mamba2-Llama3-8B (L3.1-Instr.) | 20 | 76.8 | 74.1 | 48.0 | 70.8 | 58.6 | 43.2 | - |
| Hedgehog-8B (Llama-3) | 0.04 | 77.4 | 71.1 | 40.6 | 66.5 | 54.3 | 24.2 | - |
| Llamba-8B | 12 | 80.9 | **82.5** | 54.6 | 77.6 | 73.3 | 60.0 | 69.4 |
| LoLCATs-8B | 0.04 | 81.0 | 82.4 | 54.4 | 79.1 | 73.6* | 54.9 | 67.6 |
| LoLA-8B (ours) | 0.04 | **81.6** | **82.5** | 55.4 | 79.8 | 73.6 | 57.6 | **74.9** |

---

[1] 7-9B models include Mamba (Gu & Dao, 2024), Mamba2 (Dao & Gu, 2024), RWKV-6 (Peng et al., 2024), Hawk & Griffin (De et al., 2024), Falcon Mamba (Zuo et al., 2024), RecurrentGemma (Botev et al., 2024), Mamba-in-the-Llama (Wang et al., 2024), Llamba (Bick et al., 2025), Hedgehog (Zhang et al., 2024), and LoLCATs (Zhang et al., 2025a).

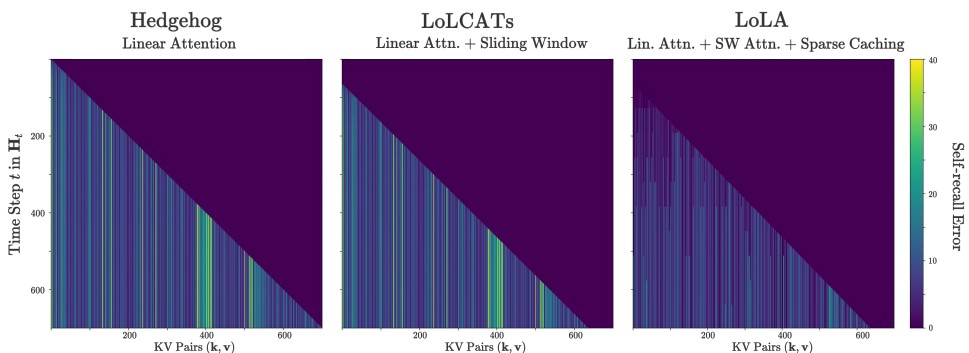

Figure 3: Visualizing memory collisions by measuring SRE for stored KV pairs.

On short context tasks such as Winogrande, we observe additional caching is not needed as only local information is required for good performance. On the other hand, we find that gaps still exist with Lambada and MMLU. Lambada requires longer context reasoning, leading to significant improvements with sparse caching. Furthermore, Bick et al. suggest that dataset selection plays a large role for MMLU performance. Though sparse caching shows significant improvement on MMLU, a more powerful distillation procedure or dataset may be needed to reach Llama's performance (Goldstein et al., 2025).

For 1B parameter models, sparse caching provides even more utility. Since the hidden state dimensionality scales with the head dimension of the base model, 1B models can face more memory collisions. In Appendix A, LoLA demonstrates state-of-the-art performance among 1B subquadratic models (such as Gated DeltaNet (Yang et al., 2024a) and Mamba), even outperforming Llama-3.2-1B on average.

## 4.3 Understanding Memory Collisions

We visualize how memory collisions occur in practice. At each time step $t$, we measure the self-recall error from equation 10 for every KV pair that is currently stored in the hidden state $\mathbf{H}_t$. We visualize the error for linear attention, sliding window + linear attention, and LoLA. We use a sliding window size of $\eta = 64$ tokens and a sparse cache size of $\lambda = 64$ when applicable.

When only using linear attention, we observe large recall errors. At early time steps, the first few KV pairs receive small errors, but quickly become larger after hidden state updates. In Appendix G, we additionally plot the relative error to better show how this occurs. These stored associative memories become corrupted and tough to recall in the future. Furthermore, difficult-to-memorize pairs are evident, illustrated as bright columns in Figure 3. The use of sliding window attention only delays the inevitable memory collisions. LoLA significantly reduces the errors for all KV pairs. Difficult-to-memorize pairs are appropriately stored in the sparse cache, shown by the zero-columns in the plot. This also prevents previously stored KV pairs from being corrupted.

## 4.4 Scoring Method Ablation

In our final experiment, we measure alternative scoring functions to understand which KV pairs should be sparsely cached. Traditional sparse attention metrics assume "unimportant" tokens are evicted entirely from the context (Singhania et al., 2024; Zaheer et al., 2020). In our setting, these assumptions are invalid as unimportant tokens are stored in low precision through linear attention.

In Table 4, we observe that storing keys with poor exponential dot product approximations underperforms the naive extension of sliding window attention. This hints that a better softmax approximation should not be the main objective for linear attention methods. Keys over-estimated by linear attention seem to be "more important" than local keys; however, all tested alternatives fall short of enabling associative recall. We also compare against traditional sparse attention ideas that use query-dependent metrics (Zhang et al., 2023;

Table 4: Ablation results for various scoring methods on S-NIAH-1 with 512 context length, $\eta = 64, \lambda = 64$. Extended details for the score calculation can be found in Appendix F.

| Importance Metric | S-NIAH-1 @ .5K | Informal Assumption for "Important" Pairs |
|---|---|---|
| $\left\| \dfrac{\phi(\boldsymbol{k})^\top \mathbf{H}}{\phi(\boldsymbol{k})^\top \boldsymbol{s}} - \boldsymbol{v} \right\|_2$ | 99.0% | *Pairs that do not align with the hidden state's prediction* |
| $(\exp(\boldsymbol{q}^\top \boldsymbol{k}) - \phi(\boldsymbol{q})^\top \phi(\boldsymbol{k}))^2$ | 11.4% | *Keys with incorrect attention weights* |
| $\lvert \exp(\boldsymbol{q}^\top \boldsymbol{k}) - \phi(\boldsymbol{q})^\top \phi(\boldsymbol{k}) \rvert$ | 20.0% | *Keys with incorrect attention weights* |
| $\dfrac{\phi(\boldsymbol{q})^\top \phi(\boldsymbol{k})}{\exp(\boldsymbol{q}^\top \boldsymbol{k})}$ | 52.0% | *Keys that Linear Attention over estimates* |
| $\exp(\boldsymbol{q}^\top \boldsymbol{k})$ | 10.6% | *Keys that are attended to during sliding window attention* |
| None, extend sliding window | 29.4% | *Most recent pairs* |

Dong et al., 2024a). Specifically, we found that a key's average similarity score, $\exp(\boldsymbol{q}^\top \boldsymbol{k})$, does not translate well in the hybrid linear attention setting. LoLA, on the other hand, benefits from its query-agnostic metric.

## 5 Conclusion

LoLA integrates linear attention with sparse caching to effectively mitigate memory collisions. By selectively retaining KV pairs that do not align with the current hidden state, LoLA enables passkey retrieval when the base model fails. Our experimental results demonstrate that targeted sparse caching substantially improves long context performance over naively increasing the sliding window size. LoLA demonstrates strong language modeling performance over other 1B or 8B subquadratic models.

**Limitations & Future Work.** While our method, in theory, can be applied on top of any sliding window + linear attention hybrid model, our experiments are limited to only two base subquadratic models (LoLCATs-Llama-3.2-1B & LoLCATs-Llama-3.1-8B). Additionally, the sparse cache carries a small overhead storage cost of $\mathcal{O}(\lambda d)$. For high-complexity, long-context tasks, we found that larger sparse cache sizes are needed to reduce interference in the hidden state. We believe that these limitations can be addressed in the future by applying LoLA's self-recall error and sparse caching to more advanced base subquadratic models. Architectures such as LaCT (Zhang et al., 2025b) or Atlas (Behrouz et al., 2025) use test-time training with nonlinear key-to-value maps, which may lead to smaller caches and be more applicable for lifelong in-context learning.

## Acknowledgments

This material is based upon work supported in part by the National Science Foundation under Grant No. NSF ECCS-2414678, the Army Research Office under Grant W911NF2410107, and the NVIDIA Corporation through the NVIDIA Academic Grant Program.

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

# A   Extended Language Modeling Results

**1B parameter model results.**   Following Section 4.2, we extend this comparison in Table 5 for various 1-2B parameter subquadratic models (Grattafiori et al., 2024; Li et al., 2023; Sun et al., 2023; Qin et al., 2024; Schlag et al., 2021; Yang et al., 2024a; Gu & Dao, 2024; Dao & Gu, 2024; Beck et al., 2024; Peng et al., 2024; Bick et al., 2024; Zhang et al., 2025a; Bick et al., 2025; Ren et al., 2025). Compared to the 8B models, we observe that sparse caching is more important in the 1B parameter regime since there exist more memory collisions. This is a direct result of a smaller hidden state dimension. The size of $\mathbf{H}_t$ scales with the head dimension of the base model. Llama-3.2 1B's head dimension is half that of Llama-3.1 8B.

We provide additional notes for the results in Table 5. Mamba and Mamba2 are popular architectures and have been trained many times with different datasets and hyperparameters. We report variations from two sources for robust results. In addition, LoLCATs demonstrated results on both Llama-3.2 1B and Phi-1.5. The model and code for reproducing LoLCATS-Phi-1.3B is not publicly available, so we could not produce Lambada scores. Similarly, we do not have LoLA results for this either. We were able to reproduce LoLCATs-Llama-1B, however, our achieved Winogrande accuracy was lower. We reported the score from the paper, 61.5%, over our reproduced 60.9%.

**Cross-teacher comparison of distilled subquadratic models.**   We gathered results from both Table 5 and Table 3 to compare language model performance relative to the teacher models. We average the performance across tasks and compute the relative average. This is calculated by dividing the model's average by the teacher model's average.

Table 5: Performance comparison across zero-shot commonsense reasoning tasks for various 1-2B parameter subquadratic models. * indicates normalized logits were reported instead.

| Model | Tokens (B) | PIQA acc ↑ | ARC-e acc ↑ | ARC-c acc_n ↑ | Hella. acc_n ↑ | Wino. acc ↑ | LMB. acc ↑ | LMB. ppl ↓ |
|---|---|---|---|---|---|---|---|---|
| **Transformers** (Bick et al., 2024; Zhang et al., 2025a) | | | | | | | | |
| Llama-3.2-1B | 9000 | 74.4 | 65.5 | 35.8 | 63.7 | 60.5 | 60.1 | - |
| Phi-1.5-1.3B | 150 | 76.6 | 75.6 | 48.0 | 62.6 | 73.4 | 53.4 | - |
| **Subquadratic:** *Pretrained from scratch on FineWeb-Edu (Yang et al., 2024a; Penedo et al., 2024)* | | | | | | | | |
| RetNet-1.3B | 100 | 70.1 | 67.3 | 33.8 | 49.2 | 54.1 | 40.5 | 17.3 |
| HGRN2-1.3B | 100 | 70.5 | 69.4 | 35.3 | 49.5 | 52.8 | 39.5 | 17.7 |
| DeltaNet-1.3B | 100 | 70.7 | 68.5 | 35.7 | 50.9 | 53.4 | 42.5 | 16.9 |
| Gated-DeltaNet-1.3B | 100 | 72.3 | 71.2 | 38.4 | 55.8 | 57.5 | 46.7 | 12.2 |
| Mamba1-1.3B | 100 | 71.3 | 69.5 | 35.4 | 52.9 | 53.0 | 44.0 | 15.1 |
| Mamba2-1.3B | 100 | 71.9 | 72.5 | 37.9 | 55.7 | 55.2 | 45.7 | 12.6 |
| **Subquadratic:** *Pretrained from scratch on various sources (Bick et al., 2024; 2025)* | | | | | | | | |
| Mamba1-1.4B | 315 | 74.2 | 65.5 | 32.8 | 59.1 | 61.5 | 64.9 | - |
| Mamba2-1.3B | 315 | 73.2 | 64.3 | 33.3 | 59.9 | 60.9 | 65.7 | - |
| xLSTM-1.4B | 300 | 74.6 | 64.3 | 32.6 | 60.9 | 60.6 | 57.8 | - |
| Finch-1.6B | 1100 | 72.6 | 64.2 | 34.1 | 57.3 | 59.4 | 66.8 | - |
| RecurrentGemma-2B | 2000 | 67.2 | 35.6 | 51.2 | 60.3 | 55.7 | 52.5 | - |
| Samba-1.3B | 100 | 72.4 | 58.2 | - | 54.7 | 55.7 | 51.7 | - |
| **Subquadratic:** *Distilled from Phi-1.5-1.3B (Bick et al., 2024; Zhang et al., 2025a)* | | | | | | | | |
| Phi-Mamba-1.5B | 3 | 75.5 | 74.0 | 44.1 | 60.2 | 71.7 | 50.1 | - |
| LoLCATs-Phi-1.3B | 0.04 | 76.9 | 77.0 | 46.9 | 62.3 | 72.7 | - | - |
| **Subquadratic:** *Distilled from Llama-3.2-1B (Bick et al., 2025; Zhang et al., 2025a)* | | | | | | | | |
| Llamba-1B | 8 | 74.0* | 69.5* | 37.2 | 61.2 | 60.6 | 48.4 | - |
| LoLCATs-Llama-1B | 0.04 | 74.6 | 63.0 | 35.1 | 63.7 | 61.5 | 53.4 | 9.3 |
| LoLA-1B (ours) | 0.04 | 76.2 | 66.2 | 36.9 | 64.1 | 60.9 | 61.9 | 5.3 |

In Table 6, LoLA outperforms other distilled model approaches. We find that LoLCATs and Llamba perform similarly overall, with LoLCATs demonstrating better token efficiency.

# B  Efficiency Analysis

**Cache Parameter Analysis.**  In this section, we analyze the efficiency of chunkwise LoLA for various sliding window ($\eta$) and sparse cache ($\lambda$) sizes. We measure the peak VRAM cost and Time-to-First-Token (TTFT) for LoLA-8B with 4K long context on an Nvidia RTX 4090. Additionally, we show how different cache parameters lead to varying performance on RULER's variable tracking task (Hsieh et al., 2024). There is no one-size-fits-all solution; balancing the trade-off between speed, memory footprint, and performance depends on the hardware budget and use-case. As a result, we provide a short guide on how to navigate this.

**Latency.**  As illustrated in Figure 4, the optimal throughput is achieved by maximizing the sliding window size and minimizing the sparse cache size. This reduces the number of chunks computed in sequential order, allowing for more intra-chunk parallelization. Small chunk sizes (i.e. $\eta = 64$) increase the update frequency for the sparse cache. We observe a significant slowdown from sparse caching in these small-chunk settings. For moderate chunk sizes, the relative slowdown from sparse caching decreases. For example with a chunk size of $\eta = 512$, we observe a TTFT of 0.57 seconds without sparse caching. This only increases to 0.63 seconds when using a sparse cache of $\lambda = 512$ tokens, $\approx 10\%$ slowdown. This contrasts with an approximate 47% slowdown at chunk size $\eta = 64$. Later in Figure 5, we show that these larger chunk sizes are practical as they only increasing the total VRAM by 0.4 GB.

Table 6: Comparison of distilled subquadratic models from different teacher models. We report the average accuracy across tasks when applicable (i.e., all scores are reported or available). We also report the relative accuracy, measured as the student average / the teacher average. Results were taken from various related works with the section header containing the sources.

| Model | Tokens (B) | PI | AE | AC | HS | WG | LB | Avg. | Rel. Avg. |
|---|---|---|---|---|---|---|---|---|---|
| **Transformers** | | | | | | | | | |
| Phi-1.5-1.3B | 150 | 76.6 | 75.6 | 48.0 | 62.6 | 73.4 | 53.4 | 64.9 | - |
| Llama-3.2-1.3B | 9000 | 74.4 | 65.5 | 35.8 | 63.7 | 60.5 | 60.1 | 60.0 | - |
| Llama-3.1-8B | 15000 | 81.1 | 81.7 | 55.1 | 79.3 | 73.9 | 73.0 | 74.0 | - |
| Llama-3.1-8B-Instr. | 15000+ | 80.8 | 81.8 | 55.2 | 79.2 | 73.9 | - | 74.0 | - |
| **Subquadratic:** *Distilled from Phi-1.5-1.3B* | | | | | | | | | |
| Phi-Mamba1.5B | 3 | 75.5 | 74.0 | 44.1 | 60.2 | 71.7 | 50.1 | 62.6 | 0.965× |
| LoLCATs-1.3B | 0.04 | 76.9 | 77.0 | 46.9 | 62.3 | 72.7 | - | - | - |
| **Subquadratic:** *Distilled from Llama-3.2-1.3B* | | | | | | | | | |
| Llamba-1.3B | 8 | 74.0* | 69.5* | 37.2 | 61.2 | 60.6 | 48.4 | 58.5 | 0.975× |
| LoLCATs-1.3B | 0.04 | 74.6 | 63.0 | 35.1 | 63.7 | 61.5 | 53.4 | 58.6 | 0.977× |
| LoLA-1.3B (ours) | 0.04 | 76.2 | 66.2 | 36.9 | 64.1 | 60.9 | 61.9 | 61.0 | 1.017× |
| **Subquadratic:** *Distilled from Llama-3.1-8B Instruct* | | | | | | | | | |
| Mamba2-Llama3-8B | 20 | 76.8 | 74.1 | 48.0 | 70.8 | 58.6 | 43.2 | 61.9 | 0.837× |
| **Subquadratic:** *Distilled from Llama-3.1-8B* | | | | | | | | | |
| Llamba-8B | 12 | 80.9 | 82.5 | 54.6 | 77.6 | 73.3 | 69.4 | 73.1 | 0.987× |
| LoLCATs-8B | 0.04 | 81.0 | 82.4 | 54.4 | 79.1 | 73.6 | 67.6 | 73.0 | 0.987× |
| LoLA-8B | 0.04 | 81.6 | 82.5 | 55.4 | 79.8 | 73.6 | 74.9 | 74.6 | 1.009× |

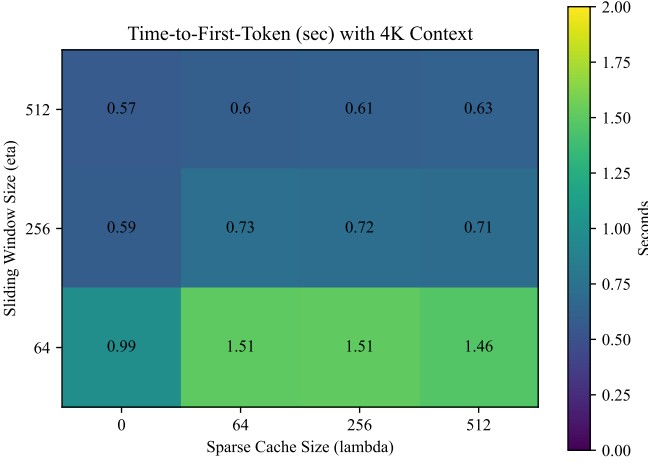

Figure 4: Measuring Time-to-First-Token for various sliding window and sparse cache sizes. This measurement is averaged across 100 trials and assumes data is already loaded into VRAM.

When $\lambda = 0$, there is no sparse caching, recovering the SWA+LA base model. Additionally, in chunkwise-computation of linear attention, the intra-chunk operations are computed in parallel. In each chunk, $\phi(\boldsymbol{q})$ and $\phi(\boldsymbol{k})$ are multiplied directly before applying a causal mask. This implies SWA and LA have similar latencies within each chunk. In the same 4K context setting as Figure 4, we recorded the TTFT for pure linear attention in PyTorch. Linear attention has a TTFT of 0.57 seconds at a chunksize of 64, and it

achieves 0.55 seconds at both chunksizes 256 and 512. For pure linear attention, all chunks can be computed in parallel, making it resistant to slowdowns at small chunksizes compared to LoLA. Across chunksizes, the peak model VRAM is 18.0 GB. Lower-level CUDA and triton implementations of linear attention are available (Yang et al., 2024c); however, we use vanilla PyTorch for an apples-to-apples comparison with LoLA's implementation. While a low-level implementation could speed up LoLA, we leave this for future work.

Table 7: Average latency of the attention layer's forward pass for various cache parameters at 4096 sequence length, 4096 embed dimension, 32 attention heads, 2 sequences in a batch. The average is computed over 250 trials.

|  | $\lambda = 0$ | $\lambda = 128$ | $\lambda = 512$ | $\lambda = 1024$ |
|---|---|---|---|---|
| $\eta = 64$ | 18.3 ms | 36.6 ms | 35.5 ms | 43.9 ms |
| $\eta = 128$ | 9.3 ms | 17.9 ms | 19.8 ms | 24.0 ms |
| $\eta = 256$ | 4.5 ms | 8.9 ms | 10.8 ms | 16.8 ms |
| $\eta = 512$ | 4.2 ms | 8.1 ms | 9.7 ms | 11.4 ms |

Additionally, in Table 7, we provide the average latency of a given layer's forward pass, compared to Figure 4 end-to-end TTFT. We observe that larger $\eta$ reduces the update frequency, diminishing the overhead cost of large $\lambda$.

Table 8: Decoding throughput after prefill for LoLA-8B across various cache parameters.

|  | $\lambda = 0$ | $\lambda = 128$ | $\lambda = 512$ | $\lambda = 1024$ |
|---|---|---|---|---|
| $\eta = 64$ | 28.2 tok/s | 27.9 tok/s | 27.9 tok/s | 27.9 tok/s |
| $\eta = 128$ | 28.1 tok/s | 28.1 tok/s | 28.1 tok/s | 28.0 tok/s |
| $\eta = 256$ | 28.1 tok/s | 28.1 tok/s | 28.1 tok/s | 28.0 tok/s |
| $\eta = 512$ | 28.1 tok/s | 28.1 tok/s | 28.1 tok/s | 28.0 tok/s |

Finally, in Table 8, we measure the decoding throughput in tokens per second for the end-to-end model. We used a 1K context length; however, once the sliding window and sparse cache are fully saturated, the context length does not impact the decoding latency. While $\eta$ is critical for speeding up prefill (Fig. 4), the cache parameters have little impact on the decoding latency, measured after prefill.

**VRAM.** Figure 5 suggests the total cache size ($\eta$ & $\lambda$) needs to be reduced to lower VRAM cost. Since window sizes greater than 64 need to be chosen for low latency from Figure 4, we observe that increasing $\eta$ to 256 or 512 is still feasible for VRAM requirements. In our implementation, the data (4K long context sequence) already exists in VRAM, so this is included in the peak VRAM measurement. Furthermore, the TTFT does not include loading this data.

**Variable Tracking Peformance.** Lastly, we show the variable tracking performance for various cache parameters in Figure 6. Variable tracking requires understanding *all* of the context. The base subquadratic model—or even extended sliding window variants—cannot perform this task without a sparse cache. As a guideline to increase general long context performance, the sparse cache size should be maximized. This mitigates memory collisions, preserving linear attention's hidden state on long sequences.

In summary, LoLA introduces a new trade-off for subquadratic models. Low VRAM and high performance can be achieved (maximize $\lambda$, minimize $\eta$), but the model will be slow. Low VRAM, fast models (minimize $\lambda$, moderate $\eta$) will not be able to perform well on long-context tasks (e.g. the base subquadratic model, LoLCATs). Finally, fast and high performing models will require a much larger memory footprint (maximize both $\lambda$ and $\eta$). This will extend the applicability of subquadratic models across various hardware platforms such as small inference chips or large training servers.

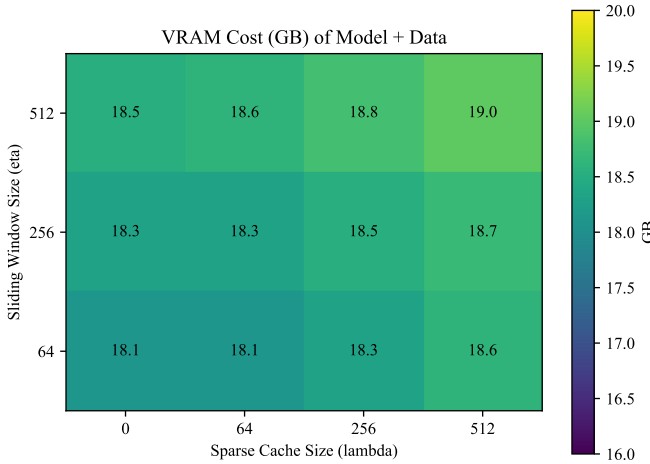

Figure 5: Measuring Peak VRAM usage for various sliding window and sparse cache sizes. This measurement includes the base model weights, the data sequence, and online activations such as KV caches.

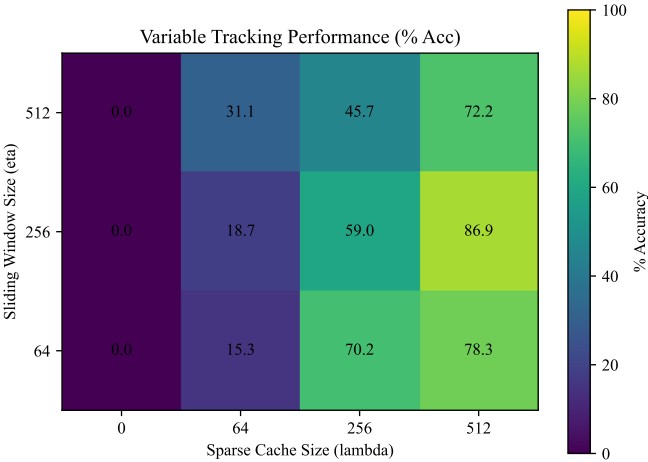

Figure 6: Cache Sizes vs. Variable Tracking performance at 4K context length from RULER (Hsieh et al., 2024).

**LoLA Cache vs. Transformers.** In Figure 7, we compare LoLA's bounded inference costs with vanilla, softmax attention. We compute the cache size as the total number of elements in all vectors and matrices stored for each attention head. For transformers, we store key and value vectors for each token in context, $t\,(d_k + d_v)$. For LoLA, we add up the elements in each of the three memory systems: sliding window cache $\eta\,(d_k + d_v)$, sparse cache $\lambda\,(d_k + d_v)$, and linear attention's hidden state $Dd_v$ & normalizing state $D$. In short context lengths, LoLA does not instantiate linear attention's hidden state or measure the SRE. LoLA is at least as efficient as softmax attention here. This means the number of elements stored for LoLA is the minimum between the number of elements in the three memory systems and the number of elements in a

traditional KV cache for a given $t$. Here, we provide both "small" and "large" cache size variants of LoLA for reference.

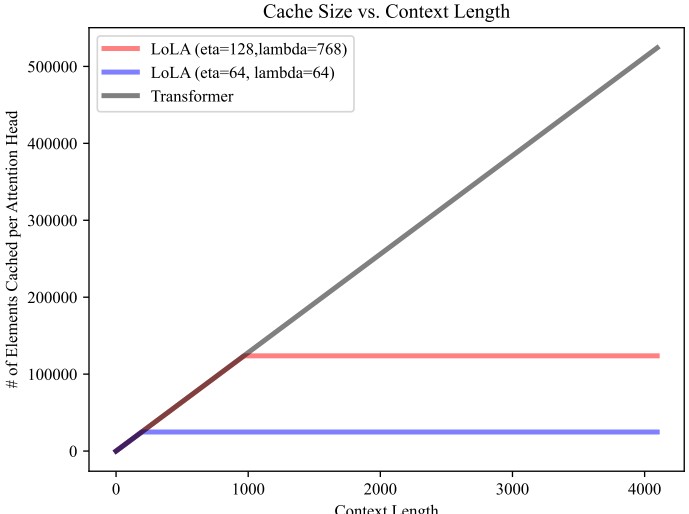

Figure 7: Cache Size vs. Context Length for LoLA and vanilla transformers.

This figure illustrates why we are interested in subquadratic models to begin with. As context scales, we must linearly increase VRAM and compute per token. For example, we observed out-of-memory errors with Llama-3.1-8B at 4K context length in our experiments. LoLA, on the other hand, can be scaled with $\eta$ and $\lambda$ to maximize performance on specific hardware. The VRAM cost is agnostic to context length, meaning this model will always be able to fit.

## C  Extended Long Context Tasks

**RULER**   To further extend our needle-in-a-haystack results from Table 1, we provide more scores in Table 9 across a greater variety of cache parameter combinations. For simplicity, we chose $\eta = \lambda$ and varied the total cache size, marked with "+". Each additional "+" doubles the total cache size (i.e baseline holds 64 tokens, + holds 128, ++ holds 256, etc.). Additionally, we provide longer sequences for S-NIAH-1 in Table 10.

In these tasks, the "haystack" is synthetically constructed with various context lengths. The first task, S-NIAH-1, uses random sentences (e.g., "The grass is green.") as the haystack, while S-NIAH-2 & 3 use essays. The needle—represented as a (word, number) pair—is placed in the haystack. At the end of the prompt, the model is tasked with returning the associated number with the special word. The first two tasks (S-NIAH-1 & 2) use a 7-digit number in the needle, and S-NIAH-3 uses a 32-digit UUID, requiring more tokens to represent the needle.

In general, extending the cache size can improve performance while still maintaining high compression rates over transformers. For even longer context lengths, LoLA's cache can easily be scaled at inference to achieve the desired recall performance. This can be seen in Table 10, where LoLA performs well up to 8K context length, which is $8\times$ longer than the sequences seen during distillation.

For large haystacks, the accuracy of LoLCATs is roughly the proportion of context that the sliding window covers. For smaller haystacks, the performance is slightly higher than that ratio since fewer pairs are stored in the hidden state. This results in fewer collisions; though of course, this does not scale. Additionally, we observe that fewer collisions exist in essay-based haystacks (S-NIAH-2 & 3), likely as a result of being more similar to the distillation data. Lastly, we observe that harder needle-in-a-haystack tasks (e.g., S-NIAH-3 with 32-digit needles) may require more sparse caching. To further extend these results, we believe training with longer sequences should yield stronger performance.

Table 9: Measuring long context recall with Needle-in-a-Haystack tasks from the RULER benchmark. We report recall accuracy for each method across different context lengths (512, 1024, etc.) for each task.

| Model | Compression Rate @ 2-4K | S-NIAH-1 | | | | S-NIAH-2 | | | S-NIAH-3 | | |
| | | .5K | 1K | 2K | 4K | .5K | 1K | 2K | .5K | 1K | 2K |
|---|---|---|---|---|---|---|---|---|---|---|---|
| **Transformer** | | | | | | | | | | | |
| Llama-3.1-8B | 1× | 100 | 100 | 100 | 100 | 100 | 100 | 100 | 100 | 100 | 100 |
| **Base Subquadratic Model** | | | | | | | | | | | |
| LoLCATs-8B | 11×-22× | 9.0 | 3.2 | 1.4 | 0.6 | 100 | 7.6 | 2.0 | 97.4 | 1.6 | 0.6 |
| **Extended at Inference** | | | | | | | | | | | |
| LoLCATs-8B+ | 6.4×-13× | 29.4 | 9.6 | 3.4 | 1.4 | 100 | 17.4 | 7.2 | 98.2 | 14.6 | 3.2 |
| LoLA-8B+ | 6.4×-13× | 99.0 | 95.4 | 79.4 | 69.4 | 100 | 39.4 | 3.0 | 99.8 | 7.4 | 1.6 |
| LoLCATs-8B++ | 4.0×-8.0× | 87.2 | 26.8 | 10.2 | 3.2 | 100 | 37.0 | 12.2 | 100 | 32.4 | 6.0 |
| LoLA-8B++ | 4.0×-8.0× | 100 | 99.6 | 96.4 | 89.6 | 100 | 98.8 | 15.0 | 99.8 | 33.4 | 9.2 |
| LoLCATs-8B+++ | 2.3×-4.6× | 100 | 65.6 | 24.6 | 8.8 | 100 | 71.8 | 21.6 | 100 | 66.0 | 10.6 |
| LoLA-8B+++ | 2.3×-4.6× | 100 | 100 | 99.6 | 97.4 | 100 | 100 | 85.4 | 99.8 | 99.8 | 27.2 |
| LoLA-8B++++ | 1.2×-2.4× | 100 | 100 | 100 | 99.0 | 100 | 100 | 100 | 100 | 100 | 100 |

Table 10: Extended Results on S-NIAH-1. Reported as [Accuracy / Compression Rate]. We evaluated performance across 500 synthetic samples on all context lengths except 16K, which used 250.

| Model | .5K | 1K | 2K | 4K | 8K | 16K |
|---|---|---|---|---|---|---|
| LoLA-8B 4+ | 100% / 1× | 100% / 1× | 100% / 1.2× | 99.0% / 2.4× | 92.2% / 4.9× | 13.6% / 9.8× |

**LongBench** Next, in Table 11 we explore long context NLP tasks from the LongBench benchmark (Bai et al., 2024), as opposed to synthetic RULER tasks. These tasks consist of Single-doc QA (MultiFieldQA (Bai et al., 2024), QasperQA (Dasigi et al., 2021)), Multi-doc QA (HotpotQA (Yang et al., 2018), 2WikiMQA (Ho et al., 2020), Musique (Trivedi et al., 2022)), and Few-shot Learning (TREC (Li & Roth, 2002), TriviaQA (Joshi et al., 2017)).

Table 11: Results on various LongBench (Bai et al., 2024) tasks such as QasperQA, MultiFieldQA, HotpotQA, 2WikiMQA, Musique, TREC, TriviaQA spanning Single-doc QA, Multi-Doc QA, & Few-shot Learning.

| Model | Tokens (B) | QQA | MFQ | HQA | 2WM | Mus | TRC | TQA | Avg |
|---|---|---|---|---|---|---|---|---|---|
| Mamba2-8B | 3500 | 25.7 | 29.3 | 30.0 | 24.6 | 11.1 | 54 | 77.77 | 36.1 |
| LoLCATs-8B | 0.04 | 1.1 | 2.5 | 1.0 | 2.1 | 0.7 | 8.2 | 1.0 | 2.4 |
| **Extended at Inference** | | | | | | | | | |
| → LoLCATs+ ($\eta = 1536, \lambda = 0$) | | 1.8 | 6.1 | 4.4 | 5.6 | 1.5 | 37.5 | 5.8 | 9.0 |
| → LoLA ($\eta = 512, \lambda = 1024$) | | 23.1 | 30.8 | 11.8 | 17.7 | 3.4 | 57.5 | 50.2 | 27.8 |

Following our findings on synthetic data, we found that the base subquadratic model (LoLCATs) cannot perform these tasks. Even if we scale the sliding window size to 1536 tokens, the extended LoLCATS+ still fails. With the same total cache size as LoLCATS+ ($\eta = 512$, $\lambda = 1024$), LoLA shows significant improvement over the base model, despite no additional training.

Similar to RULER, we cannot perform apples-to-apples comparisons with other distilled models (Bick et al., 2025; 2024; Wang et al., 2024; Zhang et al., 2024) as they do not report scores for these tasks. However, Mamba2—when pretrained from scratch on 3.5 T tokens (compared to LoLCATs 40M tokens for

distillation)—has shown very high performance here in this domain. This hints at current limitations of distilling transformers into subquadratic models.

## D    Related Work

In this section, we position LoLA within the broader landscape of subquadratic models and efficient attention mechanisms.

**Linear Attention and State Space Models (SSMs)**    State Space Models (SSMs) have emerged as powerful architectures for efficient long-range sequence modeling, offering constant memory complexity irrespective of context length. Pioneered by methods like S4 (Gu et al., 2021), recent developments include various efficient architectures such as RetNet (Sun et al., 2023) and Mamba (Gu & Dao, 2024). Concurrently, original linear attention methods have explored efficient approximations of the softmax kernel (Katharopoulos et al., 2020; Choromanski et al., 2020; Qin et al., 2022; Peng et al., 2020; Zhang et al., 2024).

**Test-Time Training.**    Test-time Regression (Wang et al., 2025) offers a unifying perspective for SSMs and linear attention. These sequence models perform online regression to fit hidden states to past context. Each state update can be interpreted as a gradient step in online SGD. This lens clarifies the roles of different mechanisms within these models.

Linear attention is the most notable form of test-time training. In the unnormalized case (i.e. without the normalization state $s_t$ normalization state) with feature maps, this performs SGD over $\mathbf{H}_t$ with the loss function $-\langle \phi(\boldsymbol{k}_t)\mathbf{H}_t, \boldsymbol{v}_t\rangle$. DeltaNet (Schlag et al., 2021) changes the loss function to the L2-Norm instead, also adding online learning rates. Gated Linear Attention (Yang et al., 2024b) and Gated DeltaNet (Yang et al., 2024a) add *gating*, which can be seen as weight decay from the optimization perspective. Lattice (Karami & Mirrokni, 2025) orthogonalizes the update. Titans (Behrouz et al., 2024) adds momentum on top of SGD. Additionally, Titans and TTT (Sun et al., 2024) provide powerful nonlinear associative maps, such as an online-learned MLP. While these have superlinear memory capacity, the update is less efficient. Even more recently, Atlas (Behrouz et al., 2025) and LaCT (Zhang et al., 2025b) build upon these MLP-based associative functions via batched (chunked) learning and muon (Jordan et al., 2024) optimization.

**The role of SRE and Test-Time Training**    LoLA's self-recall error is rooted in the same underlying principles as test-time training methods, with the goal of mapping keys to their associated values. The SRE follows a similar metric to most differentiable online objectives in these methods. For example, the $\ell^2$ loss function in DeltaNet-like models can be represented as

$$\min_{m_t} \mathcal{L}(m_t(\boldsymbol{k}_t), \boldsymbol{v}_t) = \min_{m_t} \sum_{i=1}^{t} \|m_t(\boldsymbol{k}_t) - \boldsymbol{v}_t\|_2^2 \tag{16}$$

with all tokens from 1 to t being used to compute the gradient. However, LoLA uses intra-head hybrid attention, allowing for control over which tokens are used for the online gradient (and are actually stored with $m_t$). LoLA aims to choose the subset of tokens that best fit the parametric structure of $m_t$. For example, LoLA optimizes $G_t$, the set of tokens in the sparse cache in the outer objective,

$$\min_{G_t} \min_{m_t} \sum_{i=1, \text{ st. } \boldsymbol{k}_i \notin G_t}^{t-\eta} \mathcal{L}(m_t(\boldsymbol{k}_t), \boldsymbol{v}_t), \tag{17}$$

for any test-time training loss function $\mathcal{L}$. In particular, with linear attention, LoLA estimates the "linear parts" of the KV pairs with linear attention and estimates the "nonlinear parts" with sparse attention. This outer loop minimization is not optimal; LoLA has to perform a greedy optimization for efficiency. Finally, the ideas of LoLA can be used for any parametric family of functions in test-time training.

**Comparing SRE to Gating**  In linear attention, gating can alleviate some memory collisions by de-prioritizing old information to make room for new information. This is done by using time-decaying coefficients $\gamma_i$, such that $\gamma_1 \leq \gamma_2 \leq \ldots \leq \gamma_t$, for the KV pairs in the online objective (Wang et al., 2025), with

$$\sum_i^t \gamma_i^{(t)} \| m(\boldsymbol{k}_i) - \boldsymbol{v}_i \|_2^2. \tag{18}$$

Sparse caching, on the other hand, allocates the difficult (not necessarily old) memories to sparse attention. While both methods reduce the amount of information stored in the hidden state, sparse caching keeps difficult information in full resolution. More information is preserved across the hybrid attention mechanism. While gating can improve linear attention in isolation, the linear hidden state is still limited by memory capacity (proportional to dimensionality). Nonlinear associative systems—either through hybrid attention (LoLA) or nonlinear maps—are required for superlinear capacity.

**Distilling transformers into subquadratic models.**  The cost of pretraining LLMs is the primary obstacle in finding the successor of the transformer. To address this issue, recent approaches employ knowledge distillation, transferring the capabilities of pretrained transformers into subquadratic architectures (Bick et al., 2024; Zhang et al., 2025a; Mercat et al., 2024; Bick et al., 2025; Goldstein et al., 2025). This significantly reduces training costs by recycling large pretrained models.

MOHAWK (Bick et al., 2024; 2025) demonstrated successful distillation of pretrained transformers into Mamba, maintaining competitive performance. Similarly, "Mamba in the Llama" (Wang et al., 2024) interleaves transformer and SSM blocks to retain transformer-level performance with significantly reduced inference costs. Though, this approach maintains an unbounded memory footprint due to the residual quadratic attention.

In contrast, LoLCATs use a simpler and cheaper distillation approach by using a student architecture that is more similar to the transformer. The combination of linear attention and sliding window attention significantly reduces the distillation complexity, requiring significantly fewer training tokens. LoLA directly builds on LoLCATs, leveraging its efficient distillation approach while introducing sparse caching to substantially enhance associative recall without extensive retraining.

**Sparse attention methods.**  Sparse attention methods present another orthogonal approach to reducing Transformer complexity by limiting the set of attended tokens (Nawrot et al., 2025). Methods such as Longformer (Beltagy et al., 2020) and BigBird (Zaheer et al., 2020) adopt fixed sparse patterns that incorporate sliding windows and selective global attention, efficiently capturing both local and sparse global contexts. Recent dynamic sparsification approaches, including Loki (Singhania et al., 2024) and Native Sparse Attention (NSA)(Yuan et al., 2025), employ data-dependent strategies, selectively attending to the most relevant tokens based on learned or projected keys. Native Sparse Attention, specifically, combines sparse attention with latent attention mechanisms (Liu et al., 2024), effectively approximating attention via low-rank and sparse structures.

**Hybrid approaches.**  We believe sparse attention can complement linear attention. We categorize hybrid attention approaches depending on where the attention mechanisms are mixed: intra-head, inter-head/intra-layer and inter-layer.

LoLA combines various attention techniques within the same attention head. This routes tokens between different attention implementations, preventing linear attention from being tasked with storing too much information. Other than LoLCATs (Zhang et al., 2025a) and Based (Arora et al., 2024), this intra-head design is also shared with B'MOJO (Zancato et al., 2024) and SE-Attn (Nunez et al., 2025). B'MOJO sparsely caches tokens whose outputs differ greatly from a running average of past outputs. SE-Attn instead uses a query-dependent scoring metric.

Inter-head hybrid approaches (Zhan et al., 2025; Dong et al., 2024b) combine the outputs of various attention operations within the same layer. These do not prevent memory collisions in the linear attention / SSM states; they ensemble predictions from diverse attention mechanisms. For example, HAX (Zhan et al., 2025) adds the output of sparse attention directly to the SSM output in a Mamba layer.

Inter-layer work interleaves softmax attention blocks with linear attention blocks (Ren et al., 2025; Glorioso et al., 2024; Wang et al., 2024; Zhang et al., 2025b). Since some layers use full-attention, these approaches have unbounded inference costs with respect to context length. Inter-layer and Inter-head approaches tend to provide simpler implementations, often leading to hardware-aware optimizations.

## E  Linear Attention is a Bad Low-Rank Approximation

In this section, we analyze why linear attention struggles to closely approximate softmax attention, specifically highlighting difficulties in approximating the exponential dot product kernel. We start by defining the exponential kernel's Gram matrix $\mathbf{G}$ as $\mathbf{G}_{i,j} = \exp(\boldsymbol{x}_i^\top \boldsymbol{x}_j)$ for inputs $\boldsymbol{x}_i, \boldsymbol{x}_j \in \mathbb{R}^{d_k}$. This kernel implicitly corresponds to an inner product in a potentially infinite-dimensional Hilbert space $\mathcal{H}$ via a feature map $\phi_{\exp} : \mathbb{R}^{d_k} \to \mathcal{H}$, such that

$$\exp(\boldsymbol{x}_i^\top \boldsymbol{x}_j) = \phi_{\exp}(\boldsymbol{x}_i)^\top \phi_{\exp}(\boldsymbol{x}_j). \tag{19}$$

Since explicitly working in an infinite-dimensional space $\mathcal{H}$ is infeasible, linear attention methods approximate this kernel using a finite-dimensional feature map $\phi : \mathbb{R}^{d_k} \to \mathbb{R}^D$. Consequently, linear attention approximates the Gram matrix as

$$\mathbf{G}_{i,j} \approx \widehat{\mathbf{G}}_{i,j} = \phi(\boldsymbol{x}_i)^\top \phi(\boldsymbol{x}_j), \tag{20}$$

which has a maximum rank of $D$. Ideally, $\widehat{\mathbf{G}}$ would closely approximate $\mathbf{G}$, minimizing the squared Frobenius norm error. However, linear attention's approximation error is fundamentally lower-bounded by the truncated singular value decomposition (SVD) of $\mathbf{G}$ (Eckart & Young, 1936). Specifically, for the SVD decomposition $\mathbf{G} = \mathbf{U}\boldsymbol{\Sigma}\mathbf{V}^\top$ with singular values $\sigma_i$, we have:

$$\|\mathbf{G} - \widehat{\mathbf{G}}\|_F^2 \geq \|\mathbf{G} - \mathbf{U}_D\boldsymbol{\Sigma}_D\mathbf{V}_D^\top\|_F^2 = \sum_{i=D+1}^{\text{rank}(\mathbf{G})} \sigma_i^2, \tag{21}$$

where $\mathbf{U}_D\boldsymbol{\Sigma}_D\mathbf{V}_D$ is the rank $D$ truncated SVD approximation of $\mathbf{G}$.

While the truncated SVD provides the optimal low-rank approximation, it requires the entire Gram matrix to be computed and stored, making it impractical for linear attention which demands computationally efficient, online feature mappings.

To empirically demonstrate the severity of this approximation challenge, we construct the Gram matrix under different input distributions and analyze its singular values. In our simulations, we vary both $n$ (the number of independently sampled input vectors) and $d_k$ (input vector dimensionality) and observe how they affect singular value distributions. Specifically, we draw inputs from a scaled Gaussian distribution $\boldsymbol{x}_i \sim \mathcal{N}(0, d_k^{-1/4})$ to mimic typical transformer scaling of dot products by $\sqrt{d_k}$.

Figures 8 and 9 show that applying an exponential operation to the query-key products significantly increases the rank and complexity of the resulting Gram matrix. The singular values and approximation error increase with the number of unique input vectors $n$ (see Figure 8) and the input dimension $d_k$ (see Figure 9) . Practically, in transformer architectures, head dimensions are typically modest ($d_k = 64$ for Llama-3.2 1B and $d_k = 128$ for Llama-3.1 8B). Additionally, linear attention approaches typically select feature dimensions $D$ around $2d_k$ (Zhang et al., 2024; 2025a) which can be problematic without additional sparse attention or gating mechanisms.

These experiments underscore the inherent limitation of linear attention as a softmax replacement. For an arbitarily large vocabulary size, a high dimensional hidden state is needed to truly mimic softmax. We argue that future research should exploit the inherent strengths of linear attention when it makes sense to (e.g., applying linear attention on easier-to-remember tokens), rather than attempting to replicate softmax attention in all situations.

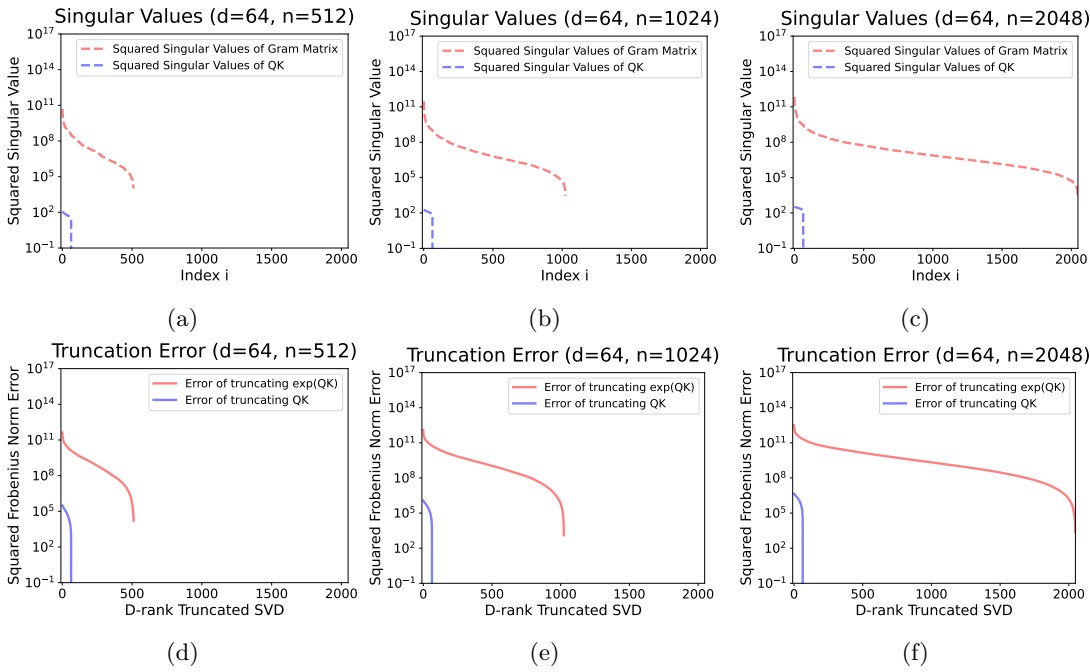

Figure 8: Visualization of singular values of the Gram matrix and minimum squared Frobenius norm error for linear attention as in equation 21. We vary the number of i.i.d. vectors $n$ used to construct the Gram matrix, but maintain the same input dimension $d$.

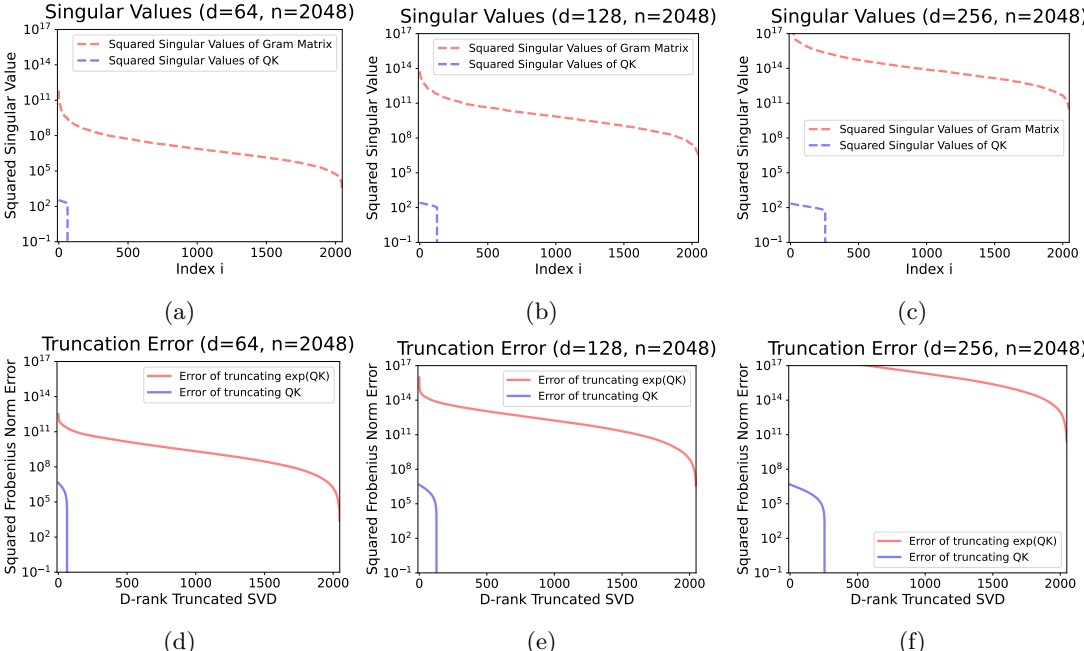

Figure 9: Visualization of singular values of the Gram matrix and minimum squared Frobenius norm error for linear attention as in equation 21. We vary the input dimension $d$ between columns in the plot.

## F   Scoring Ablation Extension

In section 4.4, we ablated different scoring approaches for the sparse cache. Here, we describe exactly how each alternative score is computed. As a reminder, LoLA is motivated by the occurrence of memory collisions in the hidden state. LoLA explicitly attempts to maintain self-recall for stored key-value pairs. An alternative perspective could be aiming for the best softmax approximation. Similar to previous kernel work (Choromanski et al., 2020), this aims to minimize the attention weight error

$$\sum_i^n \sum_j^n \left( \exp(\boldsymbol{q}_i^\top \boldsymbol{k}_j) - \phi(\boldsymbol{q}_i)^\top \phi(\boldsymbol{k}_j) \right)^2 . \tag{22}$$

A natural scoring method for this objective would be to keep the keys with the highest attention weight error. As a proxy for this approach, each key's error can be summed over all queries it sees in the sliding window with

$$\text{Score}(\boldsymbol{k}_i) = \sum_{t=i}^{i+\eta-1} \left( \exp(\boldsymbol{q}_t^\top \boldsymbol{k}_i) - \phi(\boldsymbol{q}_t)^\top \phi(\boldsymbol{k}_i) \right)^2 . \tag{23}$$

Alternatively, we can use absolute error over mean squared error instead.

From traditional sparse attention literature (Zhang et al., 2023; Dong et al., 2024a), keys that are highly attended to may be important. We compute this as

$$\text{Score}(\boldsymbol{k}_i) = \sum_{t=i}^{i+\eta-1} \exp(\boldsymbol{q}_t^\top \boldsymbol{k}_i). \tag{24}$$

From a third perspective, keys that are over-represented by linear attention's query-key interactions may seem important to cache. We compute these as

$$\text{Score}(\boldsymbol{k}_i) = \sum_{t=i}^{i+\eta-1} \frac{\phi(\boldsymbol{q}_t)^\top \phi(\boldsymbol{k}_i)}{\exp(\boldsymbol{q}_t^\top \boldsymbol{k}_i)}. \tag{25}$$

Lastly, we compare these methods against a larger sliding window.

## G   Extended Memory Collision Visualization

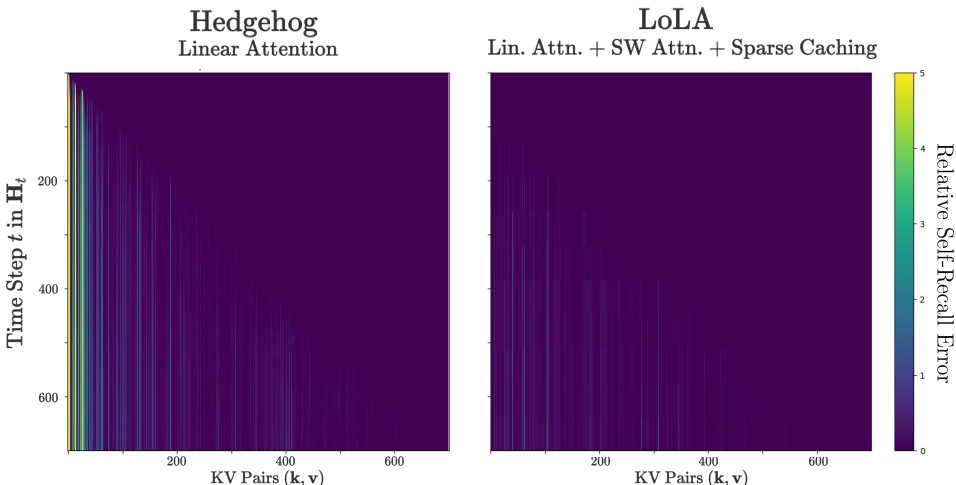

Figure 10: Visualizing the relative SRE for stored KV pairs.

In Section 4.3, we visualized how memory collisions can occur in practice. This was computed by measuring the SRE for all stored KV pairs. We found that the first few stored KV pairs do not achieve a large error when added to the hidden state. However, these quickly become corrupted after hidden state updates. This phenomenon was difficult to see in Figure 3, so we provide an additional visualization with Figure 10. Specifically, we measure the SRE of each KV pair, relative to the error of when that pair was added. For row $i$ column $j$ in the plot, the relative error is computed as

$$\left\| \frac{\phi(\boldsymbol{k}_j)^\top \mathbf{H}_i}{\phi(\boldsymbol{k}_j)^\top \boldsymbol{s}_i} - \boldsymbol{v}_j \right\|_2 - \left\| \frac{\phi(\boldsymbol{k}_j)^\top \mathbf{H}_t}{\phi(\boldsymbol{k}_j)^\top \boldsymbol{s}_t} - \boldsymbol{v}_j \right\|_2 . \tag{26}$$

where $t$ is the time pair $j$ was added to the hidden state. Thus, we have $j \leq t \leq i$.

Here, we see the first few KV pairs have high relative errors for pure linear attention. These pairs observe small SREs at early time steps, but achieve much higher SREs later. On the other hand, sparse caching actively mitigates SREs, improving associative recall for stored KV pairs.

## H    Algorithm Pseudo-Code

We provide PyTorch-like pseudo-code for the cache during prefill (update_chunk) and decoding / generation (update). Lastly, this pseudo-code does not contain any optimization tricks for ease of understanding. We also provide a simple implementation of this in PyTorch at `https://github.com/lukemcdermotttt/ LoLA-release`. This was designed to be minimal, so it can flexibly integrated into various hybrid model codebases.

```
#Simplified LoLA Cache for Decoding:
class LoLA_Cache:
    def init():
        #Cache for Sliding Window Attention
        local_cache = {keys:[], values:[]} #max size η

        #Cache for Sparse Attention
        global_cache = {keys:[], values:[]} #max size λ

        #"Cache" for Linear Attention
        H, s = zeros(D,d), zeros(D)

    #Update memory systems with a single incoming KV pair
    def update(k, v):
        eligible_keys = concat(global_cache.keys, k)
        eligible_values = concat(global_cache.values, v)

        #Predict the associated value of each key
        predicted_v = (phi(eligible_keys) @ H) / (phi(eligible_keys) @ s)
        scores = L2_norm(eligible_values - predicted_v)

        #Add min scoring KV pair to hidden state
        min_idx = argmin(scores)
        min_k = eligible_keys[min_idx]
        min_v = eligible_values[min_idx]
        H = H + phi(min_k) @ min_v.T
        s = s + phi(min_k)

        #Update Global Cache as all other KV pairs
        global_cache.keys = eligible_keys[not min_idx]
        global_cache.values = eligible_values[not min_idx]

    #Return the output associated with the query.
```

```
def attend(q):
    global_weights = exp(q @ global_cache.keys / sqrt(d) )
    local_weights = exp(q @ local_cache.keys  / sqrt(d) )

    unnormalized_attn = sum(global_weights * global_cache.values)
                      + sum(local_weights * local_cache.values)
                      + phi(q) @ h #linear attn

    normalizing_const = sum(global_weights)
                      + sum(local_weights)
                      + phi(q) @ s #linear attn

    return unnormalized_attn / normalizing_const

#Update the cache with a chunk of KV pairs
def update_chunk(k_chunk, v_chunk):

    #Score the oldest half of the sliding window
    eligible_keys = concat(global_cache.keys, local_cache.keys[: 1/2])
    eligible_values = concat(global_cache.values, local_cache.values[: 1/2])

    #Predict the associated value of each key
    predicted_v = (phi(eligible_keys) @ H) / (phi(eligible_keys) @ s)
    scores = L2_norm(eligible_values - predicted_v)

    #Keep the top-lambda scores
    threshold = get_topk(scores, k=λ)
    evicted_K = eligible_keys[scores < threshold]
    evicted_V = eligible_values[scores < threshold]
    H = H + phi(evicted_K) @ evicted_V.T
    s = s + phi(evicted_K)

    #Update Global Cache as all other KV pairs
    global_cache.keys = eligible_keys[scores >= threshold]
    global_cache.values = eligible_values[scores >= threshold]

    #Evict old SW pairs and add incoming chunk
    local_cache.keys = concat(local_cache.keys[1/2:], k_chunk)
    local_cache.values = concat(local_cache.values[1/2:], v_chunk)
```

## I  Sparse Cache & Linear Attention Ablation

In this section, we study whether the core benefit of LoLA comes from preserving memories in the linear hidden state or if sparse attention does the heavy lifting. In Table 12, we run this ablation on needle-in-a-haystack tasks from the ruler benchmark. The caches (hidden state, sparse, sliding window) are updated as normal, but we restrict which KV pairs the query can attend to. For example, if we ablate the sparse cache, the query can only attend to the hidden state and sliding window. In this example, the 512 "difficult-to-memorize" KV pairs are hidden from the query but still exist in the sparse cache. In contrast, LoLCATs add these difficult-to-memorize pairs to the hidden state; no sparse caching is used. By adding all tokens to the hidden state, LoLCATs can corrupt existing memories.

Table 12: Cache Ablations on RULER Single Needle-in-a-Haystack (SN) tasks with LoLA-8B at 2k and 4k long sequences. We also report the percent of tokens that are not attended to by the query, denoted *invisible KV pairs*, for both 2k and 4k sequence lengths. Accuracies are averaged across 50 samples for the ablation runs.

| Model | Cache Params $(\eta, \lambda)$ | Hidden State | Sparse Cache | Sliding Window | % Invisible KV Pairs at 2-4K | SN-1 (2K) | SN-1 (4K) | SN-2 (2K) | SN-3 (2K) |
|---|---|---|---|---|---|---|---|---|---|
| LoLA | (512, 512) | ✓ | ✓ | ✓ | 0% | 100 | 100 | 100 | 100 |
| LoLCATs | (1024, N/A) | ✓ | N/A | ✓ | 0% | 55.4 | 30.4 | 60 | 27.6 |
| LoLCATs | (512, N/A) | ✓ | N/A | ✓ | 0% | 24.6 | 8.8 | 21.6 | 10.6 |
| Restricted | (512, 512) | × | ✓ | ✓ | 50-75% | 14 | 6 | 2 | 0 |
| Query | (512, 512) | ✓ | × | ✓ | 25-12.5% | 26 | 14 | 24 | 10 |
| Ablation | (512, 512) | × | × | ✓ | 75-87.5% | 18 | 8 | 0 | 0 |

In this ablation, we observe that removing any of the caches can plummet performance. If the hidden state cannot be attended to (row 4), then there is not sufficient information in the sparse or sliding window cache to recall the needle in these tasks. Without the hidden state, the sparse cache does not provide additional information for sliding window attention: observe the increase in performance from row 4 to row 6.

In row 5, we allow the query to attend to the hidden state but hide the sparse cache. We still perform the same cache update rules, so these difficult-to-memorize tokens are not placed into the hidden state, unlike LoLCATs. When compared to LoLCATs with $\eta = 512$, this run uses the same sliding window size, but the hidden state is different as row 5's hidden state is missing $\lambda = 512$ KV pairs. Despite attending to fewer tokens, the row 5 ablation performs slightly better. This shows that in some circumstances, it would better to just evict the difficult-to-memorize tokens instead of adding them to the hidden state (corrupt previous memories).

Overall, this study demonstrates that all caches are important in the LoLA mechanism. By not adding difficult-to-memorize KV pairs to the hidden state, LoLA mitigates memory collisions. For best performance, the hidden state and sparse cache should both be present. Since the self-recall error is only computed within the sliding window, we must also use the sliding window cache.

