# OpenReview forum: "LoLA: Low-Rank Linear Attention with Sparse Caching"
_TMLR — Accepted by TMLR_

### Review · Reviewer_KbG1 · 2026-03-25

**Summary Of Contributions:**

Softmax attention relies on an ever-growing KV cache, which supports strong retrieval and long-context reasoning but makes extremely long contexts computationally expensive and makes lifelong in-context learning impractical. Linear attention compresses past information into a fixed-size matrix-valued hidden state, enabling constant-memory inference but weakening associative recall and long-context reasoning. Prior work has partially addressed this limitation by combining linear attention with sliding-window attention, which maintains a fixed-size KV cache of recent tokens.

The proposed method, LoLA, routes past KV pairs into three memory systems: a sliding window for recent tokens, a fixed-size KV cache for pairs that are difficult for the linear hidden state to remember, and the recurrent linear-attention state for the remaining pairs. The key technical idea is a self-recall error metric that estimates whether a key can successfully retrieve its own value from the current hidden state. This signal is used to decide which KV pairs should be stored in the KV cache. The method is appealing because it is training-free and applied at inference time on top of existing sliding-window + linear-attention models. Empirically, the paper shows large improvements on long-context retrieval benchmarks, strong gains over simply increasing the sliding-window cache at comparable cache budgets in the strongest settings, improvements on harder RULER tasks and LongBench, and modest but meaningful gains on downstream language-modeling benchmarks.

Key strengths are the clean problem formulation, the simple inference-time design, the strong mechanistic motivation around memory collisions, and the inclusion of ablations showing that self-recall error is more effective than larger sliding windows or alternative sparse-selection heuristics. A potential weakness is that the method is positioned as broadly applicable to sliding-window + linear-attention models, but the empirical validation is limited to LoLCATs.

**Additional Comments:**

1) I am not a huge fan of the + style model naming. It is undefined in the main body of the text, and makes the tables harder to parse at a glance.
2) Similarly, I find the tagline “self-recall is essential” slightly out of place and a little stronger than the evidence shown.

**Audience:**

Yes

**Audience Explanation:**

Fixed-memory long-context inference and lifelong in-context learning are important unresolved problems in machine learning, and this paper provides a simple and practically relevant contribution in that direction. More broadly, it opens up an interesting line of work on how models should make use of a fixed KV cache. I expect it to be of interest to researchers working on efficient sequence models, long-context language models, and alternatives to softmax attention.

**Broader Impact Concerns:**

I have no broader impact concerns.

**Claims And Evidence:**

Yes

**Claims Explanation:**

The paper provides strong experimental support for the main claim that targeted sparse caching improves long-context associative recall in the evaluated LoLCATs-based hybrid models. In particular, the needle-in-a-haystack results show very large gains over the base LoLCATs model, and the comparisons against larger-window LoLCATs variants make a persuasive case that the benefit is not simply coming from a larger cache budget. The extended RULER and LongBench experiments strengthen this conclusion by showing improvements on harder retrieval and long-context QA settings, while the commonsense reasoning results indicate that the method can also translate into better downstream language-modeling performance.

**Requested Changes:**

# Crticial

1) The paper claims that LoLA “pushes the Pareto front” (pp. 8, 15), but it is unclear what objective space this front is defined in. Please explicitly define the objectives, report the coordinates of LoLA and the relevant baselines in that space, and indicate which methods are actually Pareto-optimal.
2) It would be helpful to explicitly include the $\eta=0,\lambda\geq0$ regime explicitly in both the empirical baselines and the VRAM / TTFT comparisons in Appendix B. This would clarify whether any benefit is obtained from a purely sparse cache, and also how its speed and memory usage compare to plain linear attention.
3) Please explain the rationale for excluding certain natural comparison points. For example, why is LoLCATs-8B+ (1024, 0) not included in Table 1, and why is Llama-3.1-8B not included in Table 2?
4) Please explain how hyperparemeters were chosen. For example, the choice of $\eta$ and $\lambda$ in Table 2.

# Would be nice

5) Appendix B would benefit from revision. In particular, Figures 4, 5, and 6 are difficult to read and the final sentence of the first paragraph of the Latency section is difficult to follow.
6) It would be interesting to include a mechanistic interpretability study that more cleanly separates whether the core benefit comes from having the KV pairs stored in the cache or from preventing collisions in $H_t$.

---

> ### Author Response · Authors · 2026-04-29
>
> We thank the reviewer for their thorough and well-read review.
>
> ### Critical 1
>
> Our original statement:
> >"Overall, LoLA pushes the pareto front for training efficient and high-performing subquadratic LLMs." (pp. 8, 15)
>
> We will remove this off-handed statement in the next revision to avoid vague claims, as this is not essential to the paper.
>
> To be more precise in our intent, among other distilled subquadratic models in Table 6, LoLA is the most training efficient (in terms of tokens) and has the highest language modeling performance (average accuracy relative to base model). This also roughly holds in Table 3 & 5, though this is more of an apples-to-oranges comparison.  Per your request, in Table 6 LoLA-1.3B would technically be the only pareto optimal model for these axes with coordinates (.04 Billion, 1.017% rel. avg. acc) as this is both the smallest token count and highest performance.
>
> ### Critical 2
> We updated Appendix B, Latency Par. 2 to discuss pure linear attention's TTFT & VRAM (eta=0, lambda=0).
>
> For language modeling performance, Hedgehog in Table 3 provides an accuracy datapoint as it is a plain linear attention. This also uses a the same distillation approach as LoLCATs (minus the SWA lora finetuning step), making it relevant and comparable.
>
> If eta=0, lambda must be 0 because we only score new tokens when they are in the sliding window cache. This is why we used eta>0 in the TTFT comparison.
>
> If we didn't have a sliding window, we can only score the current token, meaning we are re-scoring the sparse cache every iteration (expensive). If we waited to re-score the sparse cache, then we need to put the incoming tokens somewhere. We do not want to overcrowd linear attention's hidden state & add conflicting memories, so the only other options are putting these incoming tokens into the sparse cache or some other buffer cache. This is mathematically the same as using sliding window attention.
>
> In our later response to 6), we ran an ablation showing a "purely sparse cache" (with SWA, no LA) does not perform well, as the role of sparse caching is to prevent memory collisions.
>
>
> ### Critical 3
> For our Table 1 rationale, we originally ran LoLA and LoLCATs for every configuration in the range of 64-512 total tokens in the cache to create Table 1. Then, to create Table 10 later, we ran LoLA (512,512) across a sweep of sequence lengths. After the fact, when reformatting the paper for submission, we decided to include these numbers into Table 1. We have now added LoLCATs (1024, 0) to Table 1 in our revision.
>
> We have also now added Llama-3.1-8B to Table 2 in our revision.
>
> ### Critical 4
> These parameters in Table 2 were chosen once; we did not exhaustively tune these. In Table 1, we wanted to showcase that a simple choice of eta=lambda can work. For harder tasks in Table 2, allocating more tokens in the sliding window seemed wasteful since this does not model global context. Rather than choosing (eta=512, lambda=512), we decreased eta significantly to 128. Note that dropping to 64 can slow down the TTFT as observed in Appendix B. We increased the lambda with this freed budget, but we did not want to push this towards 1-2k as we are only running on 4k long sequences. We opted for halfway in between 512 and 1024 (lambda=768). We added a brief mention of this in the revision (section 4.1).
>
> Instead of over-tuning the cache parameters in our main results, we performed a grid search for variable tracking in Appendix B as an ablation later on. For example, this grid search revealed hyperparameters with higher variable tracking performance (86.9%)  than what was found in Table 2’s experiment (85.2%).
>
> ### Would be nice 5
> We have revised the language in this section.
>
> ### Would be nice 6
> In our revision, we added an additional ablation to address this in Appendix I. Rather than letting the query attend to the kv-pairs in all forms of memory, we restricted the query to only look at specific caches. In this ablation, we see that the sparse cache does not contain important information when the linear hidden state is removed.
>
>
> ### Additional 1
> We used the "+" to denote that this augmentation is not from the original paper. In our revision we updated Table 1 Caption to define this.
>
> If you still do not like the + style naming, we will remove this or incorporate any suggestions you have to convey the information that LoLCATs+ is our stronger baseline, not from their work.
>
> ### Additional 2
> We removed this statement in our revision.

---

> > ### Comment · Reviewer_KbG1 · 2026-05-11
> >
> > Thank you for the revision. The added comparison points in Tables 1 and 2, the Appendix B discussion, and the new Appendix I ablation substantially address my main concerns.
> >
> > I have two remaining minor clarification requests. First, the Pareto-front claim still appears near Table 6, despite the response indicating that it would be removed. Given the explanation in your response, I would be happy either for this to be included with the added description or removed. Second, I believe the sparse-cache markings for LoLA and LoLCATs in Table 12 are accidentally the wrong way around.

---

> > > ### Author Response · Authors · 2026-05-12
> > >
> > > We are happy to hear that the added points, discussion, and ablation substantially addressed your concerns.
> > >
> > > Thank you for catching these errors. We have updated our draft to fix Table 12, and we removed the pareto front claim. Lastly, we will be keeping the “+” naming with this added description from the last revision.

---

### Review · Reviewer_LUcr · 2026-04-13

**Summary Of Contributions:**

While Linear Attention offers constant memory scaling, it typically struggles with high-fidelity associative recall compared to standard Transformers. The paper introduces LoLA, a training-free augmentation designed to overcome the memory capacity limitations of Linear Attention models.

**Main contribution**:
LoLA balances efficiency and high-fidelity of associative recall by partitioning Key-Value (KV) pairs across three specialized memory systems:
1. **Local Sliding Window Cache:** Holds recent tokens to maintain immediate contextual coherence.
2. **Sparse Global Cache:** Captures "difficult-to-memorize" pairs to prevent critical memory collisions.
3. **Recurrent Hidden State:** Absorbs the remaining pairs, maintaining the architecture's subquadratic speed and low memory footprint.

**Strengths**:
1. The authors developed a 'self-recall' metric to dynamically identify "difficult-to-memorize" tokens, showing it to be superior to alternative caching methods.
2. LoLA significantly improves the poor memory recall of linear attention. On pass-key retrieval tasks, LoLA recovered a performance of Llama-3.1.
3. Claims of high efficiency are supported by the results that LoLA uses a 4.6× smaller cache.
4. LoLA also demonstrates a clear performance advantage over baseline subquadratic architectures on zero-shot commonsense reasoning tasks.

**Weaknesses**
1. The paper does not define the relationship between expanding context lengths and the sparse cache size, leaving it unclear how practitioners should tune this hyperparameter as memory demands grows with context size.
2. The study is constrained by its reliance on a single baseline architecture (Llama-3.1)

**In summary**, the authors have conducted a rigorous set of experiments, and all core claims are thoroughly supported by the empirical results. While I have noted a few limitations, the paper's original approach, efficiency gains, and overall contributions significantly outweigh these concerns. I recommend this paper for acceptance. Should the authors address the highlighted limitations in their final revision, it would make my recommendation for acceptance even stronger.

**Audience:**

Yes

**Audience Explanation:**

For the last few years, alternative "subquadratic" models (like linear attention, RNNs, and state-space models) were viewed with skepticism due to their poor recall. LoLA provides strong evidence that smart memory routing (the 3-tier system and the "self-recall" metric) can make these efficient models highly competitive.

LoLA achieves near-perfect retrieval with a cache 4.6× smaller than Llama-3.1 8B. This means engineers can run smarter, longer-context models on much cheaper hardware.

**Broader Impact Concerns:**

This submission does not raise any apparent ethical concerns.

**Claims And Evidence:**

Yes

**Claims Explanation:**

The submission is supported by accurate, convincing, and clear evidence, demonstrated through the following experiments:
1. Claims of high efficiency are backed by favorable cache comparisons against standard models.
2. The architecture is rigorously tested on both long-context needle-in-a-haystack tasks and zero-shot reasoning benchmarks.
3. Performance is effectively contextualized against multiple baseline models.
4. The novel Self-Recall Error Metric is explicitly evaluated against alternative approaches.
5. Throughput ablation studies are responsibly included in the supplementary materials.

**Requested Changes:**

Overall, the study features robust ablation studies, and its core claims are well-supported. However, the inclusion of the following experiments would address existing limitations and significantly strengthen the paper:

1. Additional evaluation on how the sparse memory cache scales with increasing context sizes. An experiment demonstrating the optimal relationship between context length and the required sparse cache size would be highly valuable.
2. The method can be also validated on one additional base architecture. For example, testing LoLA on a Mistral model in addition to Llama-3.1 (similar to the cited distillation study, LolCats) would solidify the claim that this augmentation is broadly applicable.

---

> ### Author Response · Authors · 2026-04-29
>
> We thank the reviewer for their thorough review and recommendation of acceptance.
>
> ### W1 / R1
> Our work is initially motivated by fixed-hardware budget settings, where we cannot increase the modeling complexity to support the data complexity. As a result, we believe the hardware and use-case should ultimatley dictate the sparse cache size. In Appendix B we discuss how practitioners should navigate the tradeoff between latency, VRAM requirements, and task performance. Furthermore, we intended for Table 1 to demonstrate the trend between cache size and long-context performance on Needle-in-a-Haystack tasks. We agree that further experimentation on cache size vs. task performance help practioners more.
>
> ### Comment 2
> The LoLCATs Mistral model weights were not open-sourced, unlike the LoLCATs Llama weights. Rather than reproducing this distillation procedure for a different base model, we allocated our compute towards a wider variety of evaluations on Llama 8B and 1B.
>
> We agree that more base models could strengthen the empirical evidence of the paper. We have revised our paper  to include this limitation in our conclusion.

---

### Review · Reviewer_Z2F9 · 2026-04-15

**Summary Of Contributions:**

This paper introduces LoLA, an inference strategy that augments hybrid sliding-window + linear attention models with a sparse KV caching mechanism. LoLa stores past KV pairs across three complementary memory systems (i) local sliding window KV cache (ii) sparse global KV cache (iii) matrix-valued hidden state of linear attention. To manage memory allocation, authors propose to score KV pairs exiting sliding-window context with a self-recall error (SRE), measuring the retrieval error of a value when queried linear attention with its associated key. Low-error pairs are incorporated into the hidden state, while “difficult to memorize” ones are stored in a sparse cache. Selective caching mechanism reduces memory collisions and improves base model’s performance on synthetic long-context tasks from RULER benchmark, as well as on some zero-shot commonsense reasoning tasks, while maintaining fixed memory size.

**Additional Comments:**

**Questions:**

* How would the method perform without sliding-window attention (only linear attention + sparse caching)?
* Are the same KV pairs cached across all attention heads / all layers?

**Audience:**

Yes

**Audience Explanation:**

Efficient long-context modelling, KV cache compression and memory management in subquadratic sequence models are important and recently actively researched areas. A few concurrent works e.g. [1, 2, 3] address closely related problem in linear attention models and propose similar approaches based on sparse KV caching. The presence of parallel efforts demonstrates interest within the community in this topic.

[1] He, Mutian, and Philip N. Garner. "Alleviating Forgetfulness of Linear Attention by Hybrid Sparse Attention and Contextualized Learnable Token Eviction." arXiv preprint arXiv:2510.20787 (2025). \
[2] Team, MiniCPM, et al. "MiniCPM-SALA: Hybridizing Sparse and Linear Attention for Efficient Long-Context Modeling." arXiv preprint arXiv:2602.11761 (2026). \
[3] Pan, Yuqi, et al. "Scaling Linear Attention with Sparse State Expansion." arXiv preprint arXiv:2507.16577 (2025).

**Claims And Evidence:**

Yes

**Claims Explanation:**

Overall, most of the paper claims are well supported with experimental results. However, there are few concerns that weaken the overall picture:

* While the paper claims that LoLA can be applied on top of any hybrid sliding-window + linear attention model, experiments are conducted exclusively on LoLCATs distilled models. Furthermore, it remains unclear whether the poor in-context recall of the base LoLCATs model stems primarily from the linear attention architecture itself or from the short distillation context length (1024 tokens).


* Scalability beyond 8K context is questionable. Table 10 shows accuracy collapsing to 13.6% on S-NIAH-1 at 16K, which the authors acknowledge but do not resolve. This significantly limit the applicability of the proposed method to long sequences and effectively disable lifelong in-context learning.

* The paper does not compare against training-free sparse attention baselines (see e.g. [1]), that achieve selective KV retention without maintaining a fixed-size hidden state at all. Such a comparison would clarify what, if anything, the linear attention hidden state contributes beyond the sparse cache, and better justify the three-system design over simpler alternatives.

[1] Nawrot, Piotr, et al. "The sparse frontier: Sparse attention trade-offs in transformer llms." arXiv preprint arXiv:2504.17768 (2025).

**Requested Changes:**

None of the following requests are critical to my acceptance of this paper; however, I believe they would meaningfully strengthen its contributions.

* Comparison of LoLA with training-free sparse attention methods under a matched cache size.
* Investigating whether the short distillation context length (1024 tokens) isn’t a primary factor behind poor in-context recall performance of the base LoLCATs model.
* Extending evaluated context lengths on RULER benchmark tasks e.g. S-NIAH beyond 16k tokens.

Other:
* Performance of 1B models from Table 5 on RULER tasks. Especially comparison with Gated DeltaNet, which addresses the same memory collisions problem with a different mechanism.
* Release an official implementation of LoLA.
* Add pseudocode for chunkwise inference (prefill) strategy.

---

> ### Author Response · Authors · 2026-04-29
>
> We thank the reviewer for their thoughtful review and recommendation of acceptance.
>
> ### Response 1
> > While the paper
>
> We have revised our draft, including this limitation in the conclusion.
>
> ### Response 2
> > Scalability
>
> and
>
> > Extending evaluated
>
> Yes, to improve performance beyond 8k, we must increase the sparse cache size (lambda) to accommodate this. LoLA cannot prevent memory collisions when the number of conflicting KV pairs exceeds the sparse cache size. Instead of increasing lambda and evaluating beyond 16k sequence lengths, we prioritized other experiments such as a new ablation.
>
> In future work, we believe combining LoLA can be built upon nonlinear associative memory mechanisms (i.e. replace m_t in Eq. 10 with Titans/Atlas).
>
> ### Response 3
> > The paper does not...
>
> > Comparison of LoLA...
>
> The goal of this work is to develop foundational methods in the pursuit of lifelong learning, requiring model inference to maintain a bounded memory footprint.
>
> While training-free sparse attention baselines [1] are attractive methods for modern LLMs, most of these described (Block-Sparse, Vertical-Slash, FlexPrefill, Quest) do not have bounded memory cost. These query-aware methods retrieve a subset of KV pairs from the context, which requires storing the context somewhere altogether. This focuses on memory bandwidth bottlenecks instead of memory costs.
>
> In contrast,  SWA / LoLA maintain a bounded representation of past context. At any time, only a fixed amount of KV pairs are stored in full-resolution, which is important when the context length exceeds what can be stored on a device.
>
> SnapKV in theory would fit the inference requirement, as an evicted KV pair will never be used again. However, actual implementations of SnapKV do not support a bounded sized prefill as this is built for decoding. As a side note, SnapKV is roughly similar to $\exp(qk)$ in our scoring method ablation (Table 4), except we also have a linear hidden state.
>
> Furthermore, we focus on eviction-less methods, as agents should learn from all experiences. Less important knowledge and experiences should be heavily compressed or merged. Token eviction methods fundamentally fail on simple tasks that require modeling the whole context (e.g. state tracking). For a sequence of updates:
> $$x \leftarrow 5,\quad y \leftarrow x+5, \quad x \leftarrow 3, \quad z \leftarrow y+x$$
> Answering a question about $z$ requires incorporating every prior update.
> Evicting any update renders the question unanswerable.
>
>
> While current linear hidden states may be lacking, we believe compressing less-important information into a finite-sized state---rather than evicting it entirely---will be essential for future lifelong in-context learning methods. This KV memory optimization should be query-agnostic, as we are unsure which queries we will see in the future.
>
>
> ### Response 4
> > Investigating whether
>
> While using varying distillation techniques are outside the scope of this work, we added an additional ablation (Appendix I) to showcase how poor performance in the base model can be attributed to memory collisions in the hidden state. In this ablation, the sparse cache boosts the recall of the linear hidden state, but the sparse cache does not contain important information when the linear hidden state is removed.
>
> This ablation should also address "Such a comparison would clarify what, if anything, the linear attention hidden state contributes beyond the sparse cache".
>
> Memory collisions are inevitable with a finite-dimensional linear hidden state, requiring some form of hidden state management. We agree that the short distillation context length likely also contributes to poor in-context recall performance.
>
> ### Response 5
> > Performance of 1B models ...
>
> We agree that more experiments on a variety of tasks and model types will continue strengthen the work; however, we prioritized further ablations to uncover where LoLA's benefits come from.
>
> As a side note, gating is orthogonal to our approach and LoLA can be built on top of SWA + gated linear attention / deltanet models. Gating only makes room in the hidden state for recent memories. It does not prevent longterm memories from being corrupted. We discuss this in Appendix D, section "Comparing SRE to Gating".
>
> ### Response 6
> >Add pseudocode
>
> In our revision, we added pseudocode for chunkwise updates.
>
> ### Response 7
> > Release
>
> We plan to release our implementation within the next two weeks.
>
>
> ### Q1
> It wouldn't make sense to do this. We score the tokens in the sliding window before placing them in the sparse cache or hidden state. It is too expensive to re-score the sparse cache every iteration. If we wait and perform an update each chunk, we need to place these tokens in a buffer/sparse cache, which would be mathematically equivalent to having SWA.
>
> Therefore, if eta=0, then lambda should be 0 too.
>
> ### Q2
> No, each head makes its own decision as this depends on each head's linear hidden state.

---

> > ### Comment · Reviewer_Z2F9 · 2026-05-15
> > **Official Comment by Reviewer Z2F9**
> >
> > Thank you for the thoughtful response and the additional revisions. I appreciate the clarifications and new material added to the draft. Appendix I ablation directly addresses my concern about what the hidden state contributes beyond the sparse cache, and the added pseudocode is helpful.
> >
> > Maintaining my positive recommendation.

---

### Decision · Action_Editor_TXPe · 2026-05-23

**Recommendation:** Accept as is

**Audience:**

Yes

**Audience Explanation:**

Linear attention models are a new frontier in LLM training. They offer improved throughput at high sequence length at similar training costs. The paper discusses ways to improve the expressivity of linear attention models, and is thus of great interest for our community.

**Claims And Evidence:**

Yes

**Claims Explanation:**

The proposed paper discusses ways to improve the memory capacity of linear attention. The paper is well-motivated, didactic, formal, and simple to follow. The proposed approach, based on a combination of sliding-window attention, a sparse cache, and a hidden state, is effective and has convinced reviewers. The discussion with reviewers improved the paper and strengthened the discussion. The experiments (especially on RULER NIAH) clearly showcase that the proposed approach is effective.